# Decoding spatiotemporal transcriptional dynamics and epithelial fibroblast crosstalk during gastroesophageal junction development through single cell analysis

Naveen Kumar [1,2,8], Pon Ganish Prakash [2,8], Christian Wentland[2], Shilpa Mary Kurian[2], Gaurav Jethva[2], Volker Brinkmann [3], Hans-Joachim Mollenkopf [3], Tobias Krammer [4], Christophe Toussaint [4], Antoine-Emmanuel Saliba [4,5], Matthias Biebl[6], Christian Jürgensen[7], Bertram Wiedenmann [7], Thomas F. Meyer[3], Rajendra Kumar Gurumurthy [2,3] & Cindrilla Chumduri [1,2,3,7] ✉

The gastroesophageal squamocolumnar junction (GE-SCJ) is a critical tissue interface between the esophagus and stomach, with significant relevance in the pathophysiology of gastrointestinal diseases. Despite this, the molecular mechanisms underlying GE-SCJ development remain unclear. Using single-cell transcriptomics, organoids, and spatial analysis, we examine the cellular heterogeneity and spatiotemporal dynamics of GE-SCJ development from embryonic to adult mice. We identify distinct transcriptional states and signaling pathways in the epithelial and mesenchymal compartments of the esophagus and stomach during development. Fibroblast-epithelial interactions are mediated by various signaling pathways, including WNT, BMP, TGF-β, FGF, EGF, and PDGF. Our results suggest that fibroblasts predominantly send FGF and TGF-β signals to the epithelia, while epithelial cells mainly send PDGF and EGF signals to fibroblasts. We observe differences in the ligands and receptors involved in cell-cell communication between the esophagus and stomach. Our findings provide insights into the molecular mechanisms underlying GE-SCJ development and fibroblast-epithelial crosstalk involved, paving the way to elucidate mechanisms during adaptive metaplasia development and carcinogenesis.

The interplay between epithelial and stromal cells is essential to maintain tissue homeostasis, mount an effective response to local injury, and promote tissue repair[1,2]. Perturbations in this balance caused by chronic abnormal stimuli such as diet and acids promote the metaplastic adaptation of the tissue. GE-SCJ, where esophageal squamous and stomach's columnar epithelia meet, are hotspots of metaplasia (known as Barrett's esophagus (BE)) development. BE is a precursor of esophageal adenocarcinomas, whose cases have dramatically increased in the last four decades[3,4]. BE is characterized by the replacement of resident esophageal squamous epithelium with columnar or intestinal cell types usually not present in the tissue[5]. Despite this, why GE-SCJ is susceptible to BE and carcinogenesis remains unclear. Therefore, it is critical to understand the principles of GE-SCJ histogenesis and the temporal evolution of the regulatory

landscape of the GE-SCJ niches. These fundamental insights would facilitate the identification of the key regenerative mechanisms and cell-cell interaction networks that deviate from healthy homeostasis toward pathological tissue adaptations.

Advances in single-cell RNA sequencing (scRNA-seq) have enabled an unprecedented path to dissect cellular composition, heterogeneity, and the process of organogenesis. Recently, scRNA-seq studies described heterogeneity and cell identities of healthy adult esophageal or stomach epithelial cells[6,7], and the impact of aging[8], mechanical stress[9], and allergic inflammation[10] on the healthy esophageal epithelium. Further, scRNA/DNA-seq analysis of epithelial tissue deduced that BE arose from epithelial cells from submucosal glands underlying the esophagus[11] or stomach cardia[12] and its mutational landscapes[13]. Nevertheless, a deeper understanding of the evolution and delineation of squamous and columnar epithelial niches of GE-SCJ is essential and remains largely unexplored.

We and others have described the role of fibroblasts in defining the architecture of tissues, supporting the homeostasis of tissue-resident cell types, including epithelial stem cell regeneration and differentiation processes in healthy and disease states[1,14,15]. However, information on the tissue-resident fibroblasts, their heterogeneity, and their role in the specification of epithelial lineages during GE-SCJ histogenesis remains unknown.

In this study, we provided a comprehensive single-cell transcriptomic landscape of epithelia and fibroblasts and their interaction networks. We deciphered the developmental processes critical for GE-SCJ histogenesis and its homeostasis in the adult. We delineated the evolution of the embryonic bipotent primitive epithelium lining the foregut mucosa into the postnatal stratified esophagus and columnar stomach-specific epithelial lineages at GE-SCJ. We found that adult stem cells of GE-SCJ are lineage-committed and transcriptionally distinct from embryonic stem cells. By establishing organoids, lineage tracing, and spatial analysis, we demonstrate that the unique expression of the morphogenic regulators in the spatially defined fibroblasts drive the establishment of squamous and columnar epithelial niches at GE-SCJ. Further, we provide comprehensive insights into the dynamics of signaling and cell-cell communication networks between epithelial and fibroblast subgroups associated with GE-SCJ histogenesis and homeostasis. These insights highlight the importance of niche signaling, provide a valuable resource, and form a basis for understanding the largely unknown mechanisms of GE-SCJ tissue response to damage and the mechanisms behind cellular remodeling that contribute to metaplasia and cancer development.

## Results

### Single-cell map of epithelial lineage development at the GE-SCJ

The adult human esophageal mucosa is lined with stratified squamous epithelium that meets the columnar epithelium-lined stomach at the GE-SCJ (Fig. 1a). Whereas in the mouse, the esophagus opens into the stomach that comprises two regions- a stratified squamous epithelium-lined fore-stomach similar to the esophagus and columnar epithelium-lined stomach (Fig. 1a). To study the developmental process and the evolution of cellular features during GE-SCJ histogenesis, we carried out single-cell transcriptome analyses of the esophagus, GE-SCJ, and stomach tissue samples obtained from embryonic day 15 (E15), E19, newborn (pup), and adult mice. Although we expected tissue level changes during the different developmental stages of GE-SCJ, the nature of transcriptional shifts, regulatory mechanisms, and the intermediate cell types during the temporal development and GE-SCJ histogenesis is unknown. Towards this, scRNA-seq data offer a vital input source for unambiguously identifying an individual cell (or cell group) based on their transcriptional states. The uniform manifold approximation and projection (UMAP) distribution of the generated time course single-cell transcriptomes showed a clear separation of cells by developmental time at pre- and postnatal stages

(Supplementary Fig. 1a). We performed unsupervised clustering and annotated based on the expression of known lineage signatures and cell type markers. This analysis revealed the presence of squamous and columnar epithelial, stromal, endothelial, immune, and neural cell populations (Fig. 1b, Supplementary Fig. 1b). UMAP sub-clustering of epithelial cells revealed transcriptionally distinct clusters separated based on squamous and columnar lineages and reflecting their developmental state (Fig. 1c). Since esophageal epithelium at GE-SCJ is predisposed to replacement with non-resident metaplastic epithelium[16,17], we first focused on understanding the temporal evolution and establishment of epithelial lineages at the GE-SCJ during development (Fig. 1d). To identify precursor cells of squamous and columnar epithelial lineages at the GE-SCJ, pseudotime analysis using scRNA seq data was performed by reconstructing branching developmental trajectories using diffusion maps. This analysis revealed two different lineages branching out from the embryonic epithelial cells at the center from the E15 and E19 stages (Fig. 1e).

Differential expression analysis across GE-SCJ epithelial cell clusters unraveled the gene expression signature associated with embryonic precursor epithelial cells (*Sox11, Igf2, H19, Cldn6, Vcan*, and *Bex1*)[18–24] committing to either the squamous (*Trp63, Col17a1, Krt5, Krt15, Krt13, Lgals7*) or columnar (*Muc5b, Furin, Pgc, Muc6, Agr2*) epithelial lineages (Fig. 1f, g, and Supplementary Data 2). Next, we analyzed the absolute expression of embryonic precursor, squamous, and columnar epithelial marker genes in the GE-SCJ region across all time points (Fig. 1h). We found that cells expressing embryonic precursor-associated gene signatures were lost in the postnatal stages (Fig. 1h, i, Supplementary Fig. 2a, b). However, expression of *Krt7*, previously described as an exclusive marker for the residual embryonic epithelial cell at adult GE-SCJ and implicated in BE development[25,26] was observed to be expressed in cells across all the time points (Fig. 1h). These observations were further clarified by immunohistochemistry (IHC) and/or single-molecule RNA in situ hybridization (smRNA-ISH) for KRT5, P63, and KRT7 (Fig. 1j, Supplementary Fig. 2c–f). All the epithelial cells lining E13 mucosa express KRT7. However, these KRT7 cells in the esophagus and foregut region differentiate into P63 + KRT5 + cells and show reduced KRT7 expression during squamous stratification. Eventually, KRT7high cells positioned above the P63 + KRT5 + squamous epithelial cells in the esophagus and forestomach sloughed off during the E19 stage, thus visibly demarcating the KRT7low squamous and KRT7high columnar epithelial regions of the esophagus and stomach respectively in the adult stage (Fig. 1j, Supplementary Fig. 2c–f). This data shows that in the adult GE-SCJ mucosa, the columnar and squamous epithelial cells express distinct gene signatures from embryonic epithelium, indicating lineage commitment of these epithelial cells. The tree diagram delineated the epithelial differentiation steps by ordering cells based on their pseudotime values, starting from the early embryonic cells that branch into late squamous (Sq3) and columnar epithelial cells (Gland base and pit) (Fig. 1k). To identify early differentiation events, we extracted the early embryonic cell population and performed re-clustering. This revealed the presence of three subclusters within them, showing higher expression of aforementioned lineage-specific markers for squamous, columnar, and precursor populations (Fig. 1l–p, Supplementary Fig. 1c, d). The cell proportion graph further substantiates our findings that the precursor cell population was only present in the embryonic epithelial cells (at E15 and E19) and, to a very less extent, in the pup but not in the adult stage. (Fig. 1q). Similarly, the precursor cell population was restricted to embryonic stages in the esophagus and stomach epithelia (Supplementary Fig. 1e–g). Next, to understand the overall GE-SCJ epithelial characteristics, we performed combined clustering of GE-SCJ cells from all time points, revealing nine subpopulations (Jn_1 – 9) together with the projected precursor cell population that were either shared or unique during different developmental stages (Fig. 1r). Sankey analysis showed that the precursor cell population was majorly

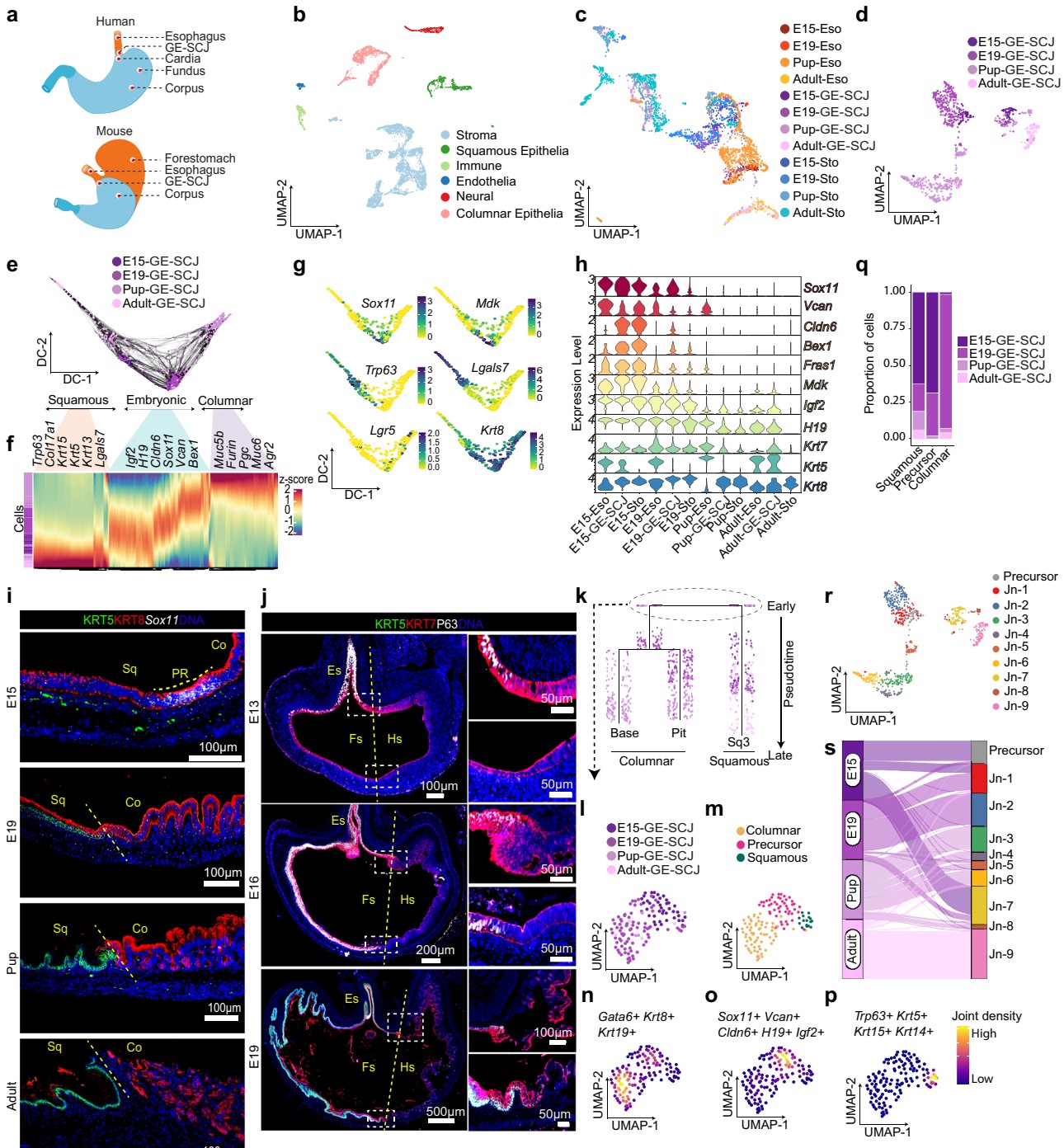

**Fig. 1 | Single-cell map of embryonic to adult epithelial cell type divergence at the GE-SCJ. a** Schematic of human and mouse adult esophagus and stomach anatomy, including GE-SCJ. **b** UMAP of scRNA-seq data of esophagus, GE-SCJ, and stomach from embryonic day 15 (E15), E19, pup, and adult mice showing six distinct cellular clusters; dots represent single cells, colored by cell types. **c** UMAP of epithelial cells, color-coded by tissue type and time point. **d** UMAP of GE-SCJ epithelial, colored by time point. **e** Diffusion map (DM) illustrates the branching differentiation of GE-SCJ epithelial cells. **f** Heatmap of differentially expressed genes (DEGs) across subclusters, with cells ordered by developmental trajectory as in (**e**). **g** Normalized expression of selected markers, visualized by DM projection as in (**e**). **h** Violin plots show expression levels of specific genes across tissues and stages. **i** smRNA-ISH and immunostaining images of mouse GE-SCJ with *Sox11* (white), KRT5 (green), KRT8 (red), and nuclei (blue). **j** Immunostained images of the mouse stomach, including distal esophagus with KRT5 (green), KRT7 (Red), P63 (white), and

nuclei (blue). Magnified view of the boxed GE-SCJ region (Right panel). Sq, Co, PR, Es, Fs, and Hs indicate squamous epithelia, columnar epithelia, precursor cell region, esophagus, forestomach, and hind stomach. Images are representative of three biological replicates in (**i**–**j**). **k** Dendrogram from URD trajectory analysis of GE-SCJ epithelial cells; each dot represents a single cell, colored by time point. Cells are ordered based on pseudotime values, starting from early at the top to late at the bottom of the tree. **l**, **m** UMAP of re-clustered GE-SCJ epithelial subpopulation positive for all selected embryonic markers (*Vcan, Igf2, Sox11,* and *H19*), colored by time point (**l**) and lineage type (**m**). **n**–**p** Joint gene-weighted density estimation of columnar (**n**), precursor (**o**), and squamous (**p**) epithelia. **q** Bar plot of epithelial types relative proportion at GE-SCJ by time point. **r** UMAP showing epithelial subclusters in combined GE-SCJ cells from E15 to adult, colored by cluster. **s** Sankey plot representing the contribution of epithelial cells from each time point to the combined GE-SCJ epithelial subclusters, as shown in (**r**).

contributed by E15 epithelial cells. In contrast, the postnatal epithelial cells majorly contributed to Jn-3–6 and 8-9 clusters (Fig. 1s).

### Evolution of squamous and columnar epithelia and their transcription factors activity during GE-SCJ histogenesis

Corroborating to scRNA seq data in Fig. 1, we observed that the adult GE-SCJ comprises two epithelial lineages, namely squamous and columnar, each characterized by lineage-specific gene expression patterns. Similar to P63 + KRT5 + and KRT7$^{high}$ expression pattern (Fig. 1j, Supplementary Fig. 2c–f), we observed that KRT8+ cells from the E13 stage differentiate to P63 + KRT5 + squamous and KRT8$^{high}$ columnar epithelia during GE-SCJ development eventually defining the adult GE-SCJ (Fig. 2a–c, Supplementary Fig. 3a–c). Furthermore, the smRNA-ISH analysis confirmed that *Krt5* and *Krt8* mRNA are specifically expressed in the adult esophagus and stomach epithelial cells, respectively (Supplementary Fig. 3d, e). Next, by inducing lineage tracing in *Krt5-CreERT2; Rosa26-tdTomato* and *Krt8-CreERT2; Rosa26-tdTomato* mice (Fig. 2d), we confirmed that the *Krt5* cells regenerate squamous epithelium of esophagus and *Krt8* cells regenerate columnar epithelium of the stomach that meet at GE-SCJ (Fig. 2e, f).

Next, we dissected the cell-type specification and subcellular differentiation within squamous and columnar lineage from the scRNA-seq data of E15, E19, pup, and adult esophagus and stomach samples. We clustered epithelial cells from the esophagus and stomach at individual time points separately (Supplementary Fig. 3f, g). E15 and E19 esophagus contains early basal stem-like epithelial sub-clusters (Sq1, Sq2), which exhibited higher expression of embryonic developmental genes such as *Sox11, Vcan, and Fras1*. Whereas the actual higher-order differentiation of epithelial cells was observed in postnatal tissues starting from the pup stage (Sq1A, Sq1B, Sq2A, Sq2B, Sq2C, Sq3). Sq1 represented the basal cell population with a remarkably higher expression of *Trp63, Krt5*, and *Col17a1*. Sq2 was positive for parabasal markers like *Jun and Fosb*, while Sq3 was positive for differentiation markers such as *Krt13, Lor*, and *Spink5* (Supplementary Figs. 3f, 4a, c). In the case of the stomach, at E15, all the epithelial cells show high proliferation and expression of embryonic developmental markers. However, two subgroups of cells showed relatively low expression of proliferation (*Mki67, Top2a*) and developmental (*Vcan*) markers, indicating the onset of differentiation of these early epithelial cells into other cell types (Neck-like and Pit-like). The presence of epithelial cell types defining the stomach gland region was evident only from E19, which contains cells expressing *Lgr5, Axin2, Chga* (Base), *Atp4a, Muc6* (Neck)*, Stmn1, Mki67* (Isthmus), *Gkn2, Tff1* (Pit) genes (Supplementary Figs. 3g, 4b, d). Cell type proportion analysis across both samples at pre- and postnatal stages showed that early embryonic columnar epithelial cells were present only in the E15 and E19 stomach samples. However, in the case of the esophagus, the basal squamous epithelium was shared at all the time points in opposition to differentiated cells that were present only during postnatal time points (Supplementary Fig. 4e). Combined clustering of epithelial cells from both esophagus and stomach across all time points revealed that the clustering of cells was not only driven by cell type but was also influenced by tissue type and developmental stages (Fig. 2g).

Pseudotime analysis of esophagus epithelial cells showed linear trajectory starting from E15, branched into two trajectories leading to differentiated states of i) E19 (Sq2) and pup (Sq2c) and ii) adult (Sq1-3) (Fig. 2h). Whereas, in the stomach, we recovered a branching tree which clearly showed the ordering of cells from embryonic to adult time points with cells from base region confined separately from cells that belong to neck and pit regions (Fig. 2i). Additionally, in the rightmost branch of the trajectory, a combination of cells mostly from E15, E19 and few from pup time points exhibited expression of early embryonic markers like *Sox11, Vcan*, while differentiated cells such as *Chga* and *Muc5ac* were found in the left trajectories mainly in pup and adult states (Supplementary Fig. 4f-i). Since scRNA-seq data represents

the cell's transcriptome at a given time, it is inferred that the embryonic differentiated cells (neck-like and pit-like), which are distinct from the differentiated adult cells on the rightmost branch, could indicate transient states and may differentiate to the adult type or likely shed off during development. Dendrogram analysis of identified cell types within the esophagus and stomach from all time points also confirmed that squamous and columnar epithelial cells were transcriptionally dissimilar (Fig. 2j). In the esophagus, basal and parabasal cells occupy separate subbranches, while highly differentiated cells (Sq2C-Pup and Sq3-Adult) appeared in a distinct subbranch, revealing transcriptional distinction between these cell types. Similarly, in the stomach, epithelial cells from the adult time point formed a separate branch, emphasizing the well-developed glandular units comprising complex cell types distinct from earlier developmental time points.

To understand the transcriptional difference and essential regulators underlying precursor cell population and stem cell compartment of the lineage-committed esophagus and stomach epithelia, we performed differential expression (DE) and transcription factors (TF) activity analysis (Fig. 2k, l, and Supplementary Data 3, 4). DE analysis showed some transcriptional similarity of precursor cell population with embryonic stem cell compartment. However, no similarity was observed with the postnatal stem cell compartment (Fig. 2k, and Supplementary Data 3). We computed TF activities based on the expression levels of their target genes. TF-target interactions were sourced from curated evidence with high confidence levels using DoRothEA[27]. This analysis revealed an overlap of cell cycle-related genes between the precursor cell population and the early-stage stem cell compartment, correlating to the higher proliferation. Columnar lineage stem cells of the stomach were enriched for the TF activities of *Gata6, Foxa1/2*, and *Hnf4a*[28–30], which were also enriched but at a lower extent in the precursor cell population, suggesting the shared identity of columnar stem cells and precursor cells. Squamous lineage-defining *Trp63, Sox2*, and *Klf5*[31] genes are only expressed in the esophageal epithelial cells. SOX2 expression was confirmed to be high in the squamous epithelium, aligning with previous findings[32], and GATA6 was highly expressed in the columnar lineage at the GE-SCJ (Fig. 2m, Supplementary Fig. S4j). GATA6 expression was confined specifically to the lower part of the stomach gland, suggesting that it might play a role in columnar stem cell maintenance and differentiation that needs to be further elucidated. In line with this, other studies have shown that GATA6 regulates intestinal epithelial proliferation, lineage maturation, and BMP repression[33–35]. Further, TFs such as *Nanog, Tead1, Prdm14, Pax5*[36,37] activity were enriched in the early-stage squamous epithelium and specific cell states of columnar epithelia (Fig. 2l, and Supplementary Data 4). However, their mechanistic role in lineage commitment within the squamous and columnar epithelia is unclear and an avenue for future research. Thus, this study provides the temporal landscape of the TF activity of epithelial stem cells during GE-SCJ development.

### Single-cell fibroblast atlas and their temporal dynamics during GE-SCJ histogenesis

To gain insights into the heterogeneity of the stromal fibroblast population, which shapes epithelial morphogenesis, we analyzed stromal cells from the pre- and postnatal esophagus, stomach, and GE-SCJ tissue regions. As a result, we identified a clear separation of stromal clusters according to pre- and postnatal developmental stages (Supplementary Fig. 5a, b). Next, to elucidate the pivotal role of underlying fibroblasts in steering the development of distinct squamous and columnar epithelia, we focused on the esophagus and stomach fibroblast cells, excluding the GE-SCJ, as it is a blend of the esophagus and stomach stromal niche (Fig. 3a). Unsupervised clustering of combined-fibroblast (C-FB) population revealed 16 transcriptionally distinct cellular subsets segregated based on tissue region and time points (Fig. 3b, Supplementary Fig. 5c). Euclidean distance measurement showed that fibroblast subpopulations from

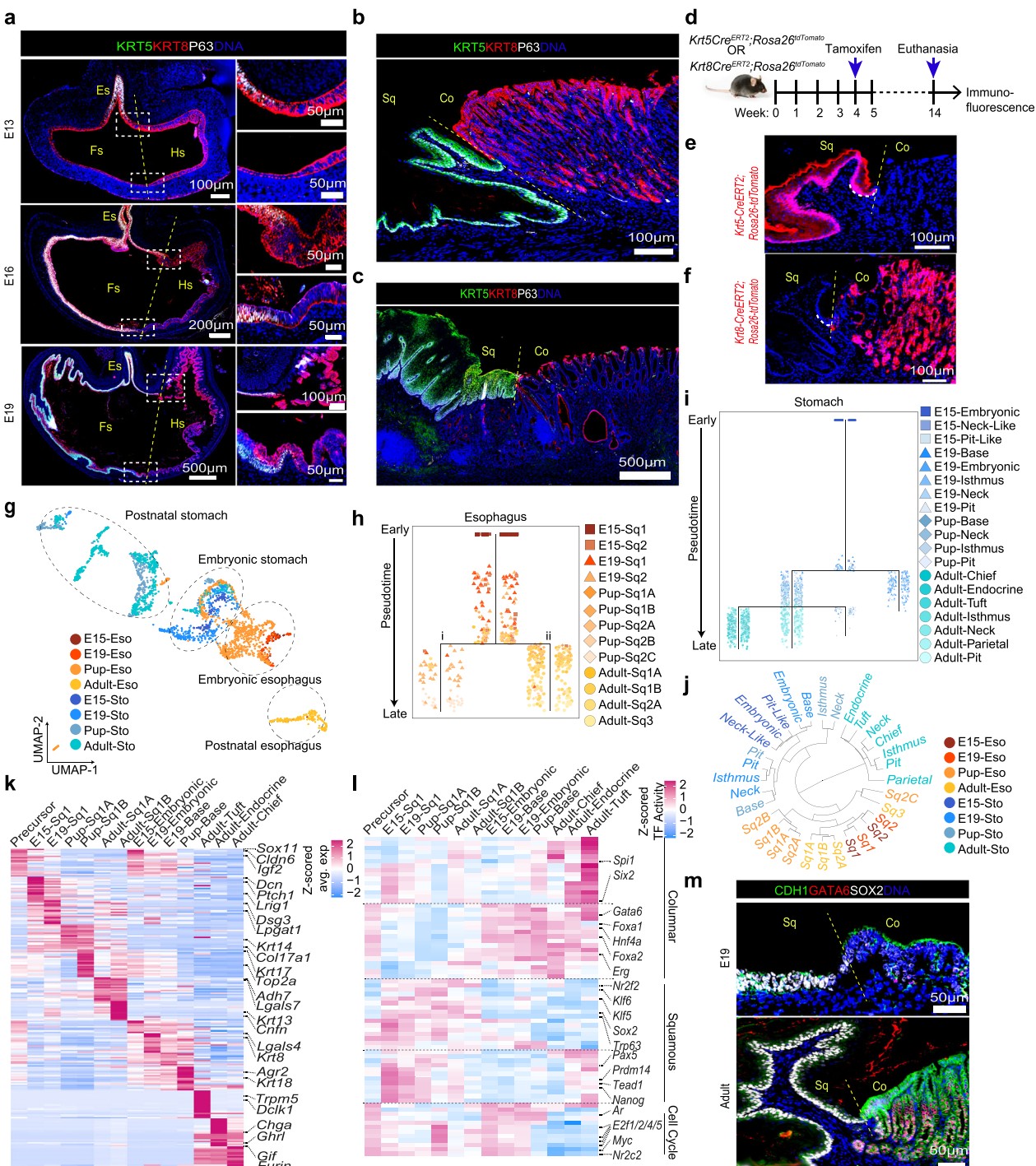

**Fig. 2 | Single-cell data and lineage tracing unravel the evolution of regulatory features of GE-SCJ development. a–c** Tiled images of the entire stomach, including distal esophagus of E13, E16, and E19 mice (**a**); GE-SCJ of the adult mouse (**b**) and human (**c**) immunostained with KRT5 (green), KRT8 (Red), P63 (white), and nuclei (blue). A magnified view of the boxed GE-SCJ regions (right panel) (**a**). **d–f** Treatment scheme for lineage tracing of mice (**d**) and tiled images of GE-SCJ tissue sections from *Krt5-Cre^{ERT2}; Rosa26-tdTomato* (**e**) or *Krt8-Cre^{ERT2}; Rosa26-tdTomato* (**f**). Nuclei (blue). The white dotted line indicates the basal cells of squamous epithelia at GE-SCJ. **g** UMAP of esophagus and stomach epithelia (excluding GE-SCJ); cells color-coded by time point. **h, i** URD differentiation tree of the esophagus (**h**) and stomach (**i**) epithelial population; each dot represents a single cell, colored by cell type. Cells ordered based on pseudotime values starting from early (top) to late (bottom). **j** Circular dendrogram indicating the similarity

between epithelial cell clusters as in (**h, i**) from both tissue types at different time points; Font color indicates time point and tissue type. **k** Heatmap showing top 20 DEG across esophagus and stomach epithelial stem cell compartments from the embryonic to adult time points; color bar denotes the z-scored mean expression range from high (deep pink) to low (blue). **l** Heatmap of 20 most variable transcription factors (TF) across epithelial stem cell compartments. The color bar depicts the scaled TF activity scores from high (deep pink) to low (blue). **m** Confocal images of the mouse GE-SCJ immunostained with CDH1 (green), GATA6 (red), SOX2 (white), and nuclei (blue). Sq, Co, Es, Fs, Hs indicate squamous epithelia, columnar epithelia, esophagus, forestomach, and hind stomach, respectively (**a–c, e, f, m**). Images are representative of three biological replicates in (**a–c, e, f, m**).

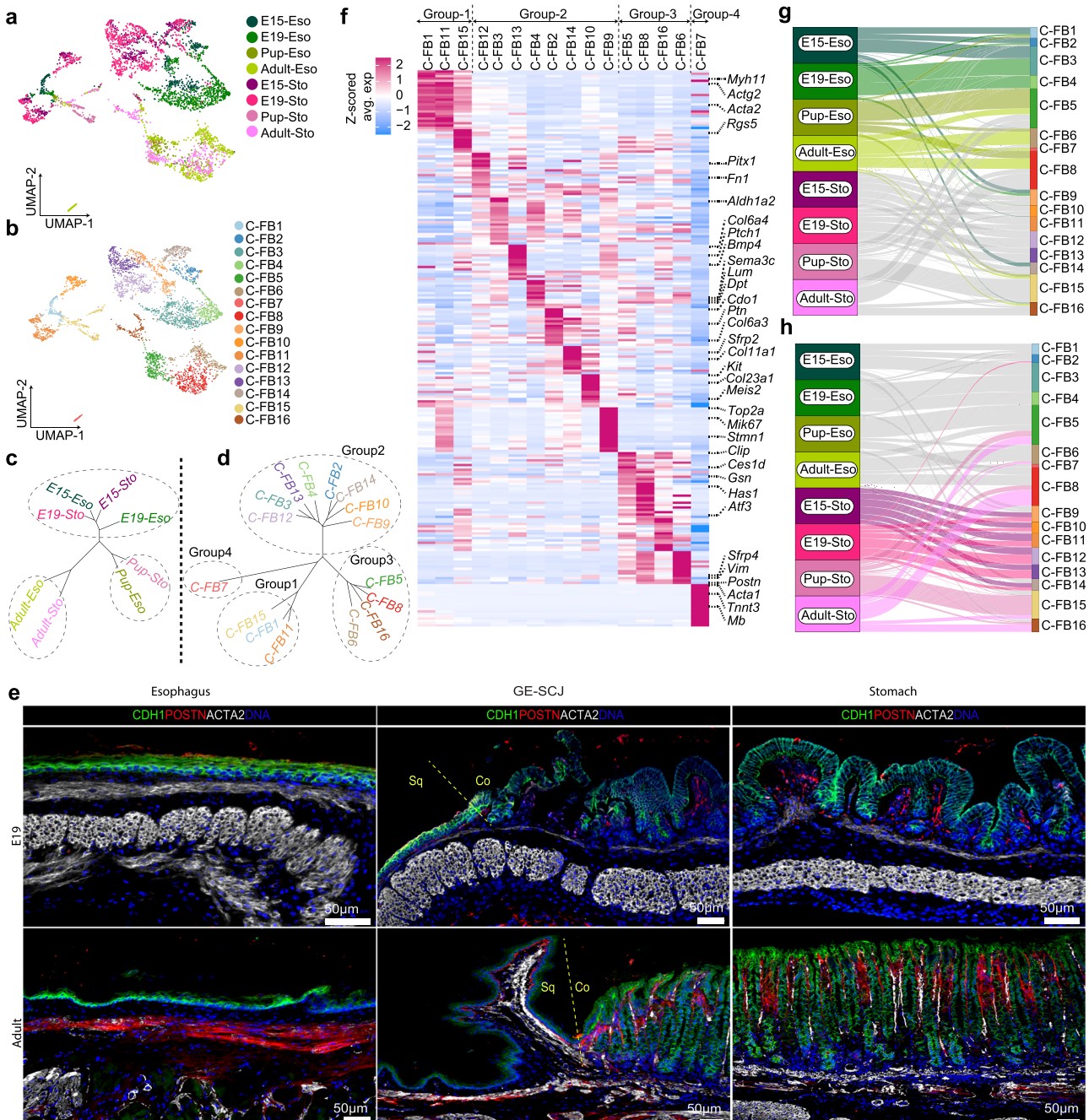

**Fig. 3 | Fibroblasts organization during GE-SCJ development. a, b** UMAP of combined fibroblast (C-FB) cell clusters from esophagus and stomach samples; colored by tissue type and time point (**a**) in shades of green and magenta, respectively, and cluster annotation (**b**). **c, d** Dendrograms highlighting the similarity between fibroblast cell clusters from esophageal and stomach tissue types at different time points (**c**) and at annotated cluster levels (**d**); font color denotes subclusters as in figures (**a, b**), respectively. **e** Tiled images of mouse esophagus, GE-SCJ, and stomach tissue sections from E19 and adult stages immunostained with CDH1 (green), POSTN (red), and ACTA2 (white) and nuclei (blue). Images are representative of three biological replicates. Sq, Co indicates squamous and columnar epithelia, respectively. **f** Heatmap of top 20 DEG across fibroblast sub-clusters as in (**b**) and subclusters were grouped as in (**d**); Color bar denotes the z-scored mean expression values ranging from high (deep pink) to low (blue). **g, h** Sankey plots highlighting the contribution of fibroblast cells from the eso-phagus (**g**) and stomach (**h**) samples at each time point to the subclusters, as shown in (**b**).

the embryonic stage grouped together and are distinct from the postnatal stromal clusters. Thus, pre- and postnatal fibroblasts possess distinct transcriptional properties (Fig. 3c). These subclusters were grouped into 4 major types based on the cell's transcriptional state similarity (Fig. 3d). Group-1 includes C-FB1, C-FB11, and C-FB15 consisting of cells from all the time points, represented by smooth muscle cells that highly expressed *Acta2, Myh11, Tagln* (Fig. 3a, d, f, Supplementary Data 5). Groups 2 and 3 expressed fibroblast marker genes

(*Col1a1, Col3a1, Dcn, Lum, Postn*) segregated into embryonic and adult fibroblasts, respectively[38]. Group 4 type fibroblasts (C-FB7) expressed muscle cell phenotypic markers such as *Acta1, Tnnt3, and Mb* and formed a distinct cluster (Fig. 3d, f Supplementary Data 5). Validation of ACTA2 and POSTN proteins in mouse E19 and Adult GE-SCJ showed the presence of two distinct Group1 and Group 2-3 fibroblast populations (Fig. 3e, Supplementary Fig. 6a, b). Among Group 2 and 3 fibroblast clusters, C-FB2-4, 10, and 16 enriched for the collagen-related

genes, suggesting their role in establishing mechanical structure during development. C-FB9 is highly enriched for the proliferation marker genes *Mki67, Top2a*, and *Stmn1*, suggesting a putative fibroblast precursor cell population in the embryonic stage. C-FB6 and C-FB8 derived from the postnatal tissue enriched for the Wingless-related integration site (WNT) inhibitor genes *Dkk2* and *Sfrp4*, indicating their role in the WNT signal modulation. The C-FB12 cluster expressed *Rgs5* and *Fn1*, previously characterized as pericyte-like cells[39]. C-FB13 exhibited strong expression of *Bmp4, Ptch1* which mediates key signaling pathways like Bone Morphogenetic Proteins (BMP) and Sonic Hedgehog (SHH), indicating a potential role in the epithelial morphogenesis during development[40,41] (Fig. 3d, f, Supplementary Data 5). We further identified the transcriptional signatures of fibroblasts specific to tissue regions (esophagus or stomach specific) and developmental stages with few markers shared over time for both esophagus and stomach (Fig. 3a, f, Supplementary Fig. 5d, f, Supplementary Data 5). The Sankey analysis highlighted the shared (C-FB2, 5, 8, 9, 14, 15, 16) or mutually exclusive (C-FB1, 3, 4, 6 for esophagus and C-FB10, 11, 12, 13 for stomach) cluster contributions of different stromal cell sub-types across the tissue during development (Fig. 3g, h). Similarly, we individually examined the distribution and heterogeneity of fibroblast types within the esophagus and stomach at all time points. We observed a clear separation of the fibroblast population between the pre- and postnatal stages, while some fibroblast states were shared across the developmental stages (Supplementary Fig. 5g–l).

Our previous study[14] shows that Wnt signaling between epithelia and stromal microenvironment plays a crucial role in dictating lineage specification. Here, we observed that *Rspo3*, a key WNT signaling agonist known for regulating stem cell regeneration[42], was expressed by a subset of fibroblasts in both esophagus and stomach (Fig. 4a, c–f). Interestingly, the proximity of *Rspo3* signals to the epithelial stem cell compartment of the esophagus and stomach differed. The average distance of the *Rspo3* signals to the epithelia is greater in the esophagus than in the stomach (Fig. 4d–f). On the contrary, *Dkk2*, a WNT inhibitory morphogen[43,44], was strongly expressed in the fibroblasts and smooth muscle cells of the esophagus with relatively low expression in the stomach (Fig. 4b, c, g–i and Supplementary Data 6). Further, expression of *Kremen1*, a receptor of DKK2[44], is observed only in the esophageal epithelial cells (Fig. 4m), suggesting the establishment of the WNT inhibitory microenvironment in the esophagus. Further lineage tracing of canonical WNT signaling target gene *Axin2*[45] in mice confirmed that esophageal epithelial cells were negative for AXIN2 lineage. In contrast, the AXIN2+ cells labeled the columnar epithelium of the stomach gland (Fig. 4n, o, Supplementary Fig. 6c). This observation was further confirmed by smRNA-ISH for *Lgr5* and *Axin2* in adult mice (Supplementary Fig. 6d–g). Together, the data revealed that the fibroblast compartment evolves concordant to the temporal development of GE-SCJ from embryonic to adult stages. The distinct sub-cell types of fibroblasts underlying the esophagus and stomach epithelia have a unique spatial organization and secrete unique location-specific morphogens. We show that the spatially defined distinct WNT fibroblast microenvironment underlying the columnar and squamous epithelia that meet at GE-SCJ plays a vital role in determining the adult GE-SCJ borders.

### Organoids of stomach and esophageal epithelium mimic in vivo distinct WNT dependency

Based on the above-observed distribution of WNT signals in the fibroblasts (Fig. 4a–o, Supplementary Fig. 6c–e), we tested the role of WNT signaling in stemness and regeneration by establishing stomach and esophageal epithelial organoids. Mouse esophageal stem cells grew into mature squamous stratified esophageal epithelial organoids in the presence and absence of WNT3a and RSPO1 (W/R) (Fig. 5a). However, they lost the stemness and growth capacity over a few passages in the presence of W/R (Fig. 5a, b, e, f). Consistently, patient-

derived esophageal cells fail to form organoids in the presence of W/R, while their absence supports the growth and differentiation into mature stratified epithelium (Fig. 5c, d). This is in contrast to previous studies that showed the culture of esophageal organoids with either the Wnt agonist R-Spondin alone[6] or in combination with a Wnt ligand[46], suggesting that Wnt signaling is dispensable for the esophageal organoid formation.

In contrast to the esophagus, and in agreement with previous studies[47,48], W/R conditioned media was essential for stomach columnar epithelial organoid growth (Fig. 5a–f). Cultured organoids maintained in vivo epithelial lineage specificity and morphology of esophagus (P63 + KRT5+) and stomach (KRT8[high], KRT7[high]), respectively (Figs. 2a–c, 5g, h, Supplementary Fig. 3a-c). A stem cell marker of the stomach, *Lgr5*, and WNT target genes *Axin2* were absent in esophagus organoids (Fig. 5m, n). Further, inhibition of endogenous WNT signaling by pan canonical and non-canonical WNT secretion inhibitor IWP2 did not influence the growth of esophageal organoids but reduced the stomach organoid growth and accelerated its differentiation with high expression of MUC5AC (Fig. 5i–l).

Next, we asked if these distinct epithelial stem cell lineages possess the plasticity to transdifferentiate with altering WNT growth factors. For this, epithelial cells from the esophagus and stomach were isolated from induced *Krt5-CreERT2;Rosa26-tdTomato* and *Krt8-CreERT2;Rosa26-tdTomato* mice, and cultured as organoids in the presence or absence of W/R media (Fig. 5o–q). Irrespective of the presence or absence of W/R esophageal stratified organoids from *Krt5-CreERT2;Rosa26-tdTomato* mice were found to be labeled, whereas matched stomach columnar organoids were not (Fig. 5p). Similarly, stomach columnar organoids from *Krt8-Cre;Rosa26-tdTomato* mice were found to be labeled, whereas matched esophageal stratified organoids were not labeled (Fig. 5q). Thus, the adult GE-SCJ consists of two committed squamous and columnar epithelial stem cells that do not transdifferentiate with the change in the WNT microenvironment. Instead, spatial WNT signaling factors play a critical role in the differential proliferation of stratified and columnar epithelia, maintaining the homeostasis of the GE-SCJ.

Further, global transcriptomic and scRNA seq analysis of the esophageal and stomach organoids corroborated the single-cell transcriptional signatures of the in vivo epithelial tissue. Microarray analysis revealed that among 34393 unique probes, encompassing protein-coding genes and long non-coding RNAs, 8030 genes were differentially regulated between columnar and squamous epithelium (Supplementary Fig. 7a, Supplementary Data 7). Gene ontology terms associated with the differentially expressed genes between the esophagus and stomach organoids showed enrichment of distinct pathways specific to the epithelial types (Supplementary Fig. 7b and Supplementary Data 8). Pathways related to epidermal cell development, keratinocyte differentiation, transcription and translation, and regulation of cell-cell adhesion were highly enriched in the esophageal epithelial cells. In the stomach epithelial cells, metabolic and catabolic processes related to lipids, fatty acids, and ion transport were enriched. While WNT signaling was critical in regulating GE-SCJ homeostasis, our analysis revealed that columnar epithelial cells were enriched for the canonical WNT beta-catenin and non-canonical WNT/Ca[2+] pathway genes. In contrast, squamous epithelial cells were enriched for the non-canonical WNT/planar cell polarity (PCP) pathway genes (Supplementary Fig. 7c).

Further, scRNA seq analysis revealed the heterogeneity and subcellular composition of columnar and squamous epithelial cells of gastroesophageal organoids. We categorized cells from stomach (ST) organoids into two major clusters (ST-Co1, ST-Co2 and the squamous epithelial cells of esophageal (ES) organoids were segregated into five unique clusters (Sq1, Sq2A, Sq2B, Sq3A and Sq3B) (Fig. 5r). The UMAP recapitulates the differentiation stages of the columnar stomach and stratified esophageal epithelial cells. The ST-Co1 subcluster was

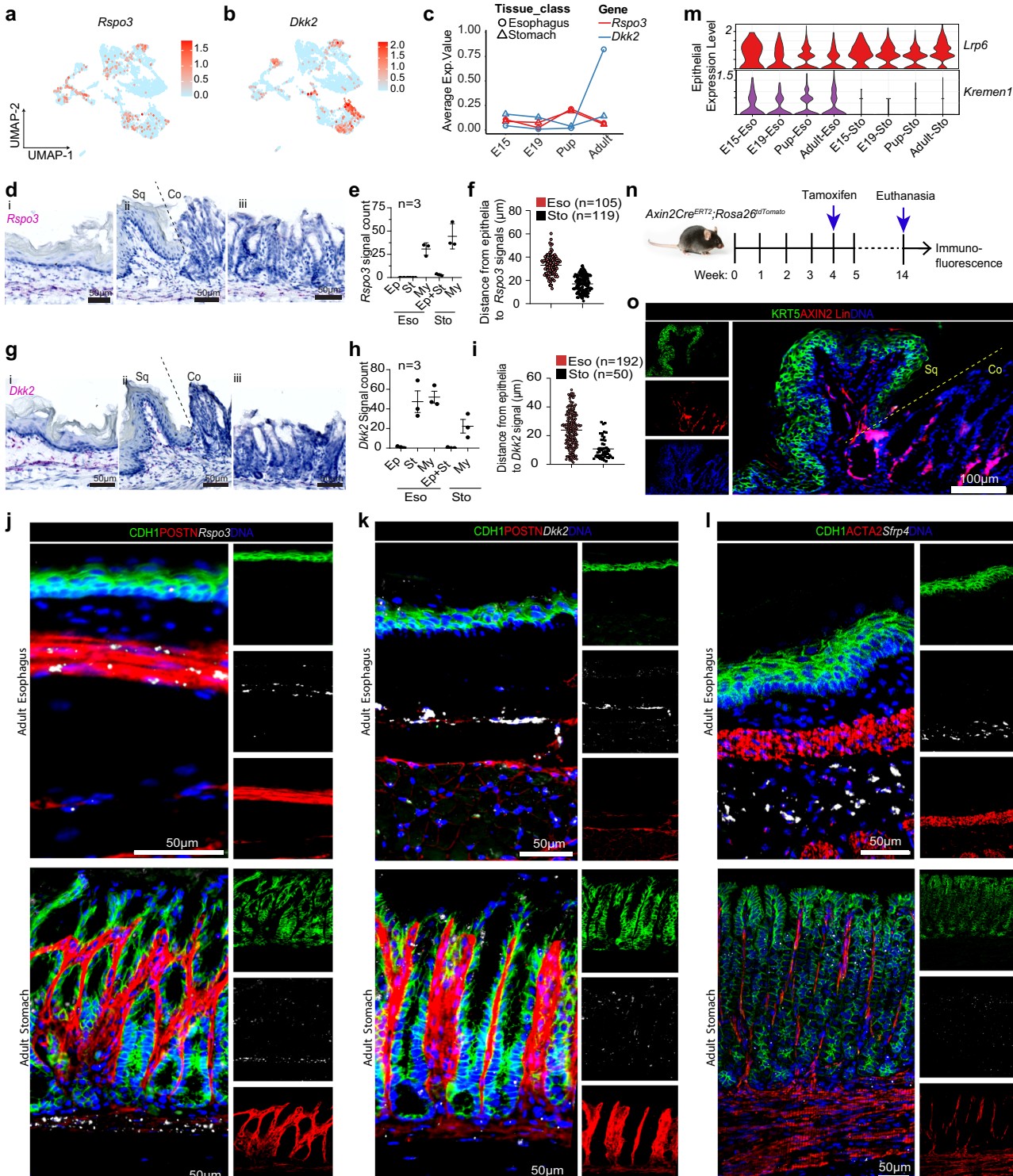

**Fig. 4 | WNT microenvironment in regulation of gastroesophageal tissues.**
**a**, **b** Feature plots showing normalized expression levels of markers *Rspo3* (**a**) and
*Dkk2* (**b**) within fibroblast cells. **c** Trend plots depict the changes associated with
mean expression levels of the selected markers over time, as in (**a**, **b**). Line color
denotes genes, and point shapes represent tissue type. **d**–**i** smRNA-ISH images of
the WNT pathway genes *Rspo3* (**d**) and *Dkk2* (**g**) in the mouse esophagus tissue (i),
GE-SCJ (ii), and stomach glands (iii). Nuclei (blue). Quantification of *Rspo3* (**e**) and
*Dkk2* (**h**) signal counts in epithelia (Ep), stroma (St), and myofibroblast (My) in the
mouse GE-SCJ tissue regions and distance (μm) from epithelia to *Rspo3* (**f**) and *Dkk2*
(**i**) signal. Data are mean +/- SEM (**e**, **f**, and **h**, **i**). *n* = number of signal count and their
distance to epithelia (**f**, **i**) from three non-overlapping 100 μm² regions of

esophagus and stomach tissues. **j**–**l** Confocal images of adult mouse esophagus and
stomach tissue sections immunostained for CDH1 (green), POSTN (red), and ACTA2
(red) and smRNA-ISH for *Rspo3* (white), *Dkk2* (white) and *Sfrp4* (white) as indicated.
**m** Violin plot showing the normalized gene expression values of *Lrp6* and *Kremen1*
from embryonic to adult time points at different tissue regions. **n** Scheme for
lineage tracing of mice expressing *Axin2-Cre^ERT2^/Rosa26-tdTomato*. **o** Tiled images of
GE-SCJ sections from *Axin2-Cre^ERT2^/Rosa26-tdTomato* mice co-immunostained for
KRT5 (green), AXIN2 lineage traced cells marked by Tdtomato (red), and nuclei
(blue). Sq, Co indicates squamous and columnar epithelia, respectively. Images are
representative of three biological replicates in (**d**, **g**, **j**–**l**, **o**). For (**e**, **f**, and **h**, **i**), source
data are provided as a Source Data file.

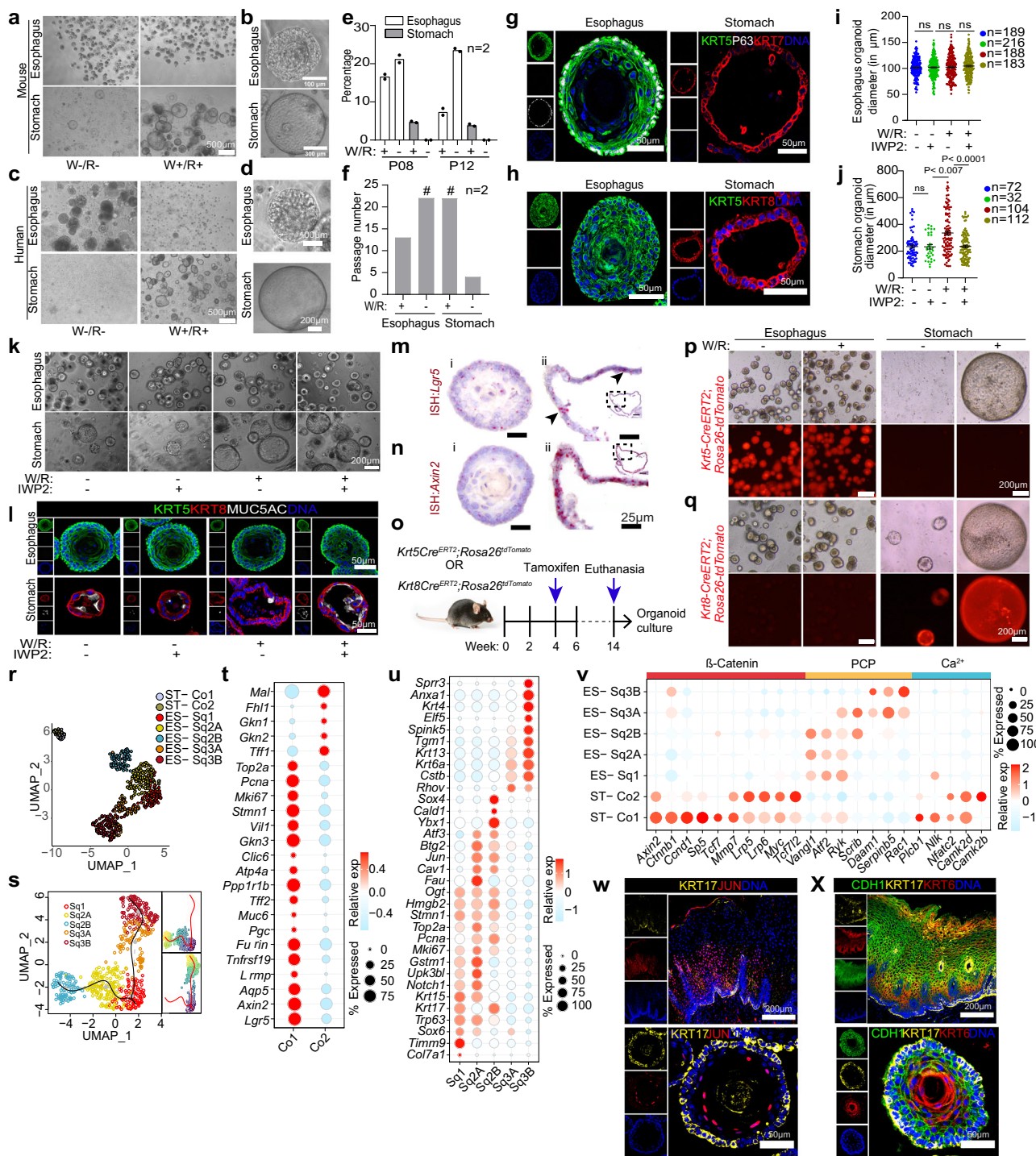

enriched for the expression of well-known stomach stem cell markers *Lgr5, Aqp5,* and *Axin2* with high levels of *Pgc, Muc6, Gkn3,* and *Atp4a* expression, which are key markers of cells present in the neck and isthmus region. These cells also expressed high levels of proliferation markers, including *Mki67, Pcna, Top2a,* and *Stmn1.* The second sub-cluster, ST-Co2, comprises mostly pit cells of the stomach gland, which expressed high levels of *Gkn1, Gkn2,* and *Tff1* (Fig. 5r, t). The esophageal subcluster Sq1 expressed *Col7a1, Timm9, Trp63, Stmn1,* and *Krt17,* representing the stratified epithelium's basal cells. The Sq2A sub-cluster consists of transient proliferating cells expressing *Mki67, Top2a, Pcna, Fau, Gstm1, Jun,* and *Upk3bl.* The subcluster Sq2B was enriched for *Atf3, Cav1, Ybx1, Cald1,* and *Sox4, while* Sq3A and Sq3B subclusters exhibited differentiation-associated gene markers such as

*Rhov, Krt6a, Krt13, Anxa1, Tgm1, Spink5, Gsta5, Sprr3* and *Elf5* (Fig. 5r, u, Supplementary Fig. 7d–g). Similar to our bulk transcriptomic data (Supplementary Fig. 7c), we further identified the distinct expression patterns of the canonical and non-canonical WNT signaling genes in subpopulations of the columnar and esophageal epithelium from the scRNA seq data (Fig. 5v).

Since little is known about the esophageal epithelial differentiation trajectories in vitro, we performed a pseudo-temporal reconstruction of the lineage using slingshot[49]. We show two distinct trajectories, all originating from the basal stem cell compartment of Sq1, differentiating into distinct sub-lineages Sq2 and Sq3 (Fig. 5s). Further, by immunostaining, we spatially located the cell types in scRNA seq data that express KRT17, JUN, and KRT6 in human and

**Fig. 5 | Distinct WNT signaling dependency for esophageal and stomach epithelial organoid growth validates the in vivo WNT microenvironment.**
**a–d** Bright-field images of the mouse (**a**, **b**) and human (**c**, **d**) esophageal and stomach organoids grown in the presence or absence of WNT3A (W) and R-spondin1 (R). **b**, **d** Higher magnification of (**a**, **c**). **e**, **f** Percentage of organoid formation (**e**) and long-term passaging (**f**) from esophagus and stomach under indicated conditions and passages (P); data derived from two biological replicates (n = 2). '#' indicates organoids can be passaged beyond the stated number.
**g**, **h** Images of mouse esophageal and stomach organoid immunolabeled for KRT5 (green), KRT7 (Red), P63 (white), KRT8 (Red), nuclei (blue). **i**, **j** Organoid diameter measurement from mouse esophagus (**i**) and stomach (**j**) grown in indicated media. n = number of organoids measured. Data are representative of three biological replicates. Data are mean +/- SEM; statistical significance was calculated using a two-sided t-test, P-values as indicated. **k**, **l** Bright-field (**k**) and confocal images showing KRT5 (green), KRT8 (red), MUC5AC (white), and nuclei in blue (**l**).

**m**, **n** smRNA-ISH images of *Lgr5* (**m**) and *Axin2* (**n**) in mouse esophagus (**i**) and stomach organoids with inset images (ii). *Lgr5*-highlighted in arrowhead (**m**-ii).
**o–q** Scheme for lineage tracing of mice (**o**). Organoids cultured from cells lineage traced for KRT5 (**p**) and KRT8 (**q**) in indicated media. **r** UMAP showing cellular subclusters of esophageal and stomach epithelial organoids. Cells colored by cluster (ST, stomach; ES, esophagus; Co, Columnar epithelia; Sq, squamous epithelia). **s** Pseudotime trajectories in esophagus epithelial subclusters. **t–v** Dot plot depicting relative gene expression for stomach (**t**) and esophagus (**u**) epithelial subclusters for canonical and non-canonical WNT pathway (**v**). Circle size denotes percentage of cells expressing a gene; color represents the scaled mean expression level from high (red) to low (blue) (**t–v**). **w, x** Images of human tissue (upper panel) and mouse esophagus organoids (lower panel), immunostained for KRT17 (yellow), JUN (red), KRT6 (red), CDH1 (green) and nuclei (blue). Images are representative of three biological replicates in (**a–d**, **g-h**, **k–n**, **p-q**, **w-x**). For (**e**, **f**, and **i**, **j**), source data is provided as a Source Data file.

mouse tissue and organoids, revealing three major subtypes, KRT17 + / JUN- basal stem cells KRT17 + /JUN+ parabasal cells and KRT6+ differentiated cells (Fig. 5w–x). Thus, organoids reflect the in vivo epithelial heterogeneity and illustrate the differential impact of WNT signaling on gastroesophageal epithelial stem cell regeneration and differentiation dynamics.

## Spatio-temporal alteration in epithelial and fibroblast signaling patterns

Our approach by employing tissue and organoid models and transcriptome analyses at both global and single-cell levels indicated that the spatial signaling factors are crucial in dictating the squamocolumnar epithelial homeostasis in GE-SCJ. Hence, to gain insights into the pathways and uncover the molecular regulatory networks between epithelial and fibroblast cell populations during GE-SCJ development, we performed gene set enrichment analysis (GSEA) using scRNA-seq data. We identified key signaling pathways differentially enriched between tissue types and time points (Fig. 6a, and Supplementary Data 9). Pathways such as bile acid and fatty acid metabolism were enriched in the stomach epithelia. While MYC target genes were enriched in esophagus and stomach epithelia, they gradually decreased towards the adult stage, suggesting an overall reduction in cell proliferation as higher-order differentiation proceeded with development. Interestingly, stroma from both esophagus and stomach exhibited strong enrichment for PI3K- FGFR1 cascade, Platelet-Derived Growth Factor (PDGF) signaling, and myogenesis. The hallmark of inflammatory response was more upregulated in both adult tissue stromal regions, and the hallmark of complement was highly enriched in the esophagus stromal cells, suggesting the presence of activated fibroblast[50].

However, the enrichment results did not reveal information regarding the directionality and temporal dynamics of these signaling pathways. Therefore, we scrutinized for alterations in signaling patterns and their strengths between embryonic and adult stages using comparative CellChat[51] analysis. In order to mitigate the complexity of cellular interactions and their interpretation, we designated E19 and adult mice as representatives for the pre- and postnatal stages, respectively, and were used for the interaction study. We found that many pathways, such as Laminin and FN1, were enriched during both the pre- and postnatal stages of the esophagus, while pathways including MK, NCAM, and VCAM were more enriched in the prenatal esophagus; Transforming Growth Factor Beta (TGF-β), Fibroblast Growth Factor (FGF), and Chemokine (C-X-C motif) Ligand (CXCL) were more enriched in the postnatal esophagus (Supplementary Fig. 8a). Interestingly, in case of stomach, majority of the pathways showed more enrichment during the pre-natal phase (Supplementary Fig. 8a).

Next, we identified the patterns for incoming, outgoing (Supplementary Fig. 8b, c), and overall signaling associated with epithelial and

fibroblast cells (Fig. 6b, c). In our analysis, 'incoming' or 'receiver' signals refer to the communication received by a cell population through expressed receptors. Conversely, 'outgoing' or 'sender' signals pertain to the communication initiated by a cell population, typically through the expression of ligands. Our analysis indicated that fibroblasts predominantly served as the signaling senders during the epithelial-fibroblast interplay in the esophagus and stomach (Supplementary Fig. 8b, c). For Instance, in the esophagus, the Notch pathway has consistently stronger incoming signals in the epithelium compared to fibroblasts at both E19 and adult stages. At the E19 stage, fibroblasts predominantly exhibit outgoing Notch signals, whereas in adult tissues, epithelial cells emerge as the primary source. This pattern indicates that epithelial cells function as receivers of Notch signals across both examined stages. In contrast, fibroblasts transition from being predominant senders at E19 to a less active signaling role in adults (Supplementary Fig. 8b). This observation aligns with our earlier study, emphasizing the significance of basal squamous epithelial stem cells as the primary source of outgoing Notch signal and differentiated cells as the receivers contributing to stratification[14].

Overall interactions for cell adhesion signaling pathways, including collagen, THBS, Laminin, and FN1, were higher in fibroblast cells of both pre-and postnatal stages, whereas NCAM, VCAM, and OCLN were found higher only in prenatal fibroblasts. Further, TGF-β signaling was highly expressed in fibroblasts of the prenatal stomach, while in postnatal phase, it was more active in the esophagus. When compared between the esophagus and stomach, the signaling strength for BMP, non-canonical WNT (ncWNT), NOTCH, WNT, and FGF was retained at a similar level during esophagus development, whereas in the stomach, signaling was predominant at the early stage (Fig. 6b, c). These results provide a comprehensive overview of the evolution of organ-specific epithelial-stromal signaling, which regulates several biological processes and homing of tissue-resident cells during the histogenesis of GE-SCJ[52,53].

Next, we checked for the sources and targets of signaling involved in the development associated pathways such as WNT, BMP, TGF-β, Insulin-like Growth Factor (IGF), FGF, NOTCH, SHH, and PDGF. We manually collected and curated key ligands (L), receptors (R), and positive and negative modulators (M) for each pathway (from publicly available literature together with the Kyoto Encyclopedia of Genes and Genomes (KEGG) database) and assessed their mRNA expression level across all epithelial and stromal subclusters of E19 and adult esophagus and stomach samples. We used the individual time point-based subclustered fibroblasts (Supplementary Fig. 8d, e) and epithelial cells of both the esophagus and stomach for analysis (Supplementary Fig. 3f, g). This comprehensive analysis unraveled a detailed expression pattern of L-R-M across various epithelial and stromal subclusters, offering insights into the intricate network of epithelial-fibroblast communication during the GE-SCJ development (Fig. 6d). BMP pathway genes were expressed relatively more in the fibroblasts than

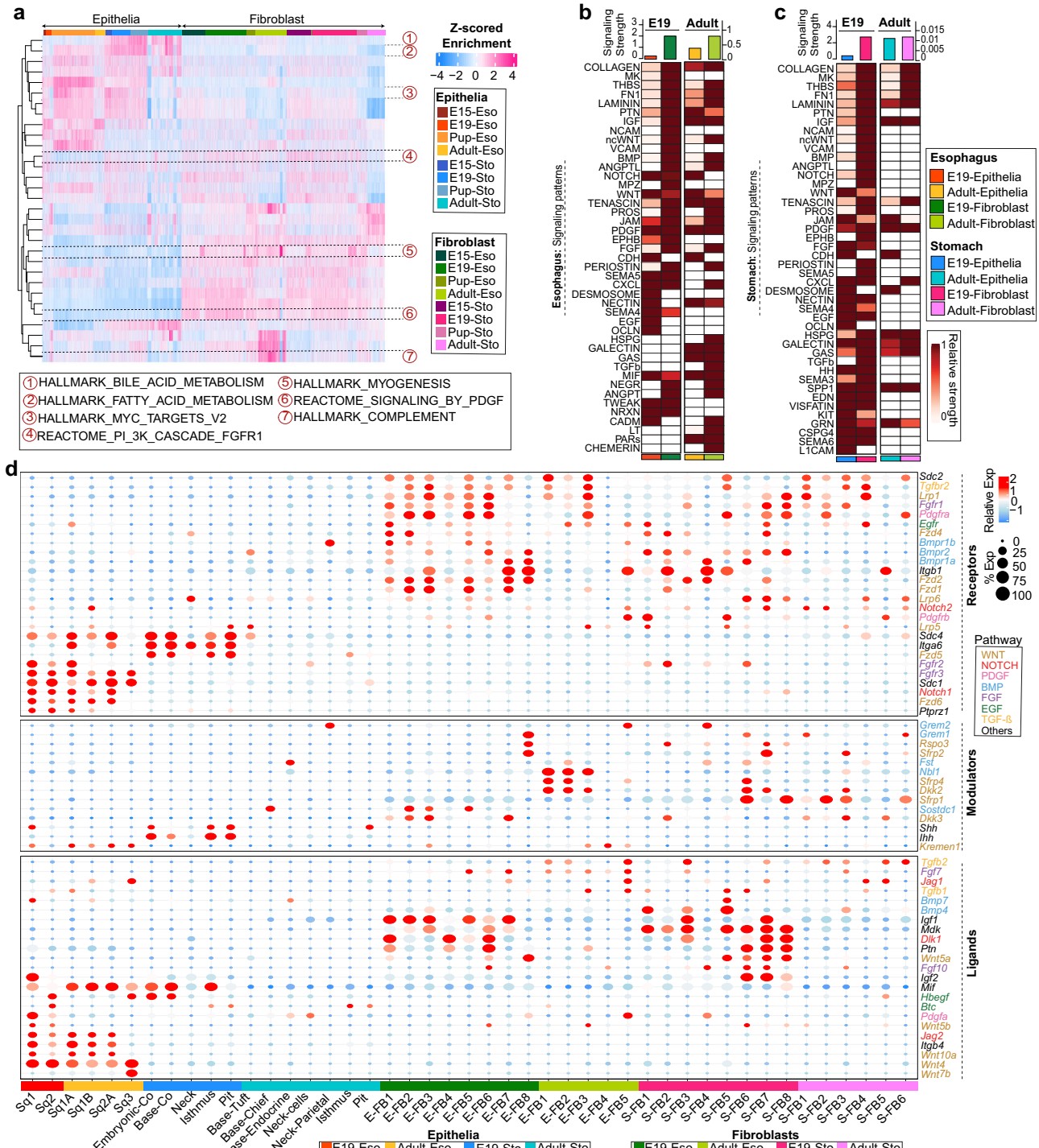

**Fig. 6 | Spatio-temporal alteration in epithelial and fibroblast signaling patterns of pre- and postnatal gastroesophageal tissues. a** Heatmap of gene set enrichment scores of fibroblasts and epithelial cells of esophagus and stomach from embryonic to adult time points with specific pathways highlighted; column represents individual cells colored by tissue type and time point; colors in the scale bar denotes the z-scored enrichment values ranging from high (deep pink) to low (blue). **b**, **c** Heatmap comparing the overall (aggregated both incoming and outgoing) signaling patterns associated with both fibroblast and epithelial compartments in the esophagus (**b**) and stomach (**c**) between E19 and adult time points. The color bar denotes the relative signaling strength (row-scaled values) of a pathway across cell types and time points. The relative strength of a pathway is calculated by normalizing each row of values to fall within the range 0-1 and depicted as low (white) to high (dark brown). Colored bar plot on top depicts the total signaling strength of a particular cell type by summarizing all pathways in the heatmap. **d** Dot plot showing the expression levels of ligands, receptors, and modulators associated with key signaling pathways in both fibroblasts and the epithelial subpopulation of esophagus and stomach at E19 and adult stages. Dot size represents the percentage of cells expressing a particular gene; the color bar indicates the intensity of scaled mean expression levels ranging from high (red) to low (blue). Genes are color-coded based on the signaling pathways to which they belong.

epithelial cells throughout development. Other pathway genes, such as *Igf1, Mdk*, and *Ptn*, were highly expressed in the fibroblasts of both esophagus and stomach during the prenatal stage. The distinct expression profiles of FGF ligands in fibroblasts, with *Fgf7* highly expressed in the esophagus and *Fgf10* in the stomach, suggest a regulatory role in the GE-SCJ. The expression patterns of *Fgf7* and *Fgf10* align with their requirement for esophageal[13] and stomach[48,54] epithelium, as evidenced by organoid studies[6,32]—nonetheless, their precise contribution to GE-SCJ development remains to be elucidated. Hedgehog signaling genes *Ihh* and *Shh* were expressed in high levels in stomach epithelia during the prenatal stage, while receptors like *Notch1, Sdc1, Fgfr2*, and *Fgfr3* were expressed in high levels in esophageal epithelial cells. WNT ligand genes *Wnt4, Wnt5b, Wnt7b*, and *Wnt10a* were strongly expressed only by squamous epithelia. In particular, *Wnt4* was highly expressed among all esophageal epithelial subclusters, indicating its role in epithelial-stromal interaction, proliferation, and differentiation in the stratified epithelium[55]. WNT receptor *Fzd6* plays a significant role in the PCP pathway during development and is an inhibitor of cWNT signaling specifically expressed at a higher level in the esophagus epithelial subclusters[56,57]. The known ncWNT ligand *Wnt5b* was briefly expressed in the early esophagus, while *Wnt5a*[58] was highly expressed in the fibroblasts of the stomach. The Wnt inhibitors *Dkk2* and *Sfrp4* expressions were restricted to the fibroblasts of the adult esophagus (Fig. 6d). Taken together, our data reveal differential pathway enrichment and alterations in the signaling patterns between squamous and columnar niches governing GE-SCJ development and homeostasis.

### Decoding fibroblast-epithelial crosstalk at ligand-receptor level during GE-SCJ development

To better understand epithelial-fibroblast interactions, we analyzed signaling interactions based on ligand-receptor pairs between epithelia and fibroblasts at a subcluster level. This analysis retrieved unknown additional information on autocrine and paracrine signaling. We identified significant ligand-receptor pairs by combining differential expression analysis with cell-cell communication analysis. Our results revealed that pathways such as WNT, BMP, TGF-β, Epidermal Growth Factor (EGF), FGF, and PDGF were among the significant ones. Overall, cell-cell interaction showed fibroblasts predominantly sent FGF and TGF-β signals to the epithelia. In comparison, PDGF and EGF signals were sent predominantly from epithelial cells to fibroblasts. The BMP and WNT signals act in both autocrine and paracrine manner in both epithelia and fibroblasts. However, the type of ligands and receptors involved varied between the esophagus and stomach (Fig. 7a–c, Supplementary Fig. 9a–c). Further, we investigated the direction of signaling involving significant ligands identified from our cell-cell interactions (Fig. 7a–c, Supplementary Fig. 9a–c, left panel) together with ligands and receptor expression dynamics across developmental time points in both the stomach and esophagus (Fig. 7a–c, Supplementary Fig. 9a–c, right panel). Interestingly, *Tgfb2* and *Fgf7* expression levels increased over time in esophageal fibroblasts, whereas *Pdgfa/b/c* and *Hbegf* expression exhibited a declining trend over time in the epithelia of both tissues (Supplementary Data 10).

Further, the inferred significant L-R pairs for BMP, TGF-β, FGF, EGF, cWNT, ncWNT, and PDGF-mediated communications between epithelia and fibroblasts were visualized using a chord diagram (Fig. 7d–f, Supplementary Fig. 9d–f). FGF signaling takes place in both autocrine and paracrine manner, where signals are usually sent by the fibroblasts and directed towards epithelial and fibroblast cells in both the esophagus and stomach (Supplementary Fig. 9d). In the case of EGF signaling, different ligands were expressed by the differentiated squamous epithelial cells and stomach epithelial cells (Supplementary Fig. 9e). These ligands interact in both autocrine and paracrine settings by binding to either *Egfr* or *Egfr-Erbb2* receptor pair, implying that epithelia are the signaling source and signals were directed either back

to epithelia or towards fibroblasts in both esophagus and stomach. Our ligand-receptor analysis of WNT signaling revealed that esophageal cells express *Wnt4, Wnt10a, Wnt7b, Wnt5a*, and *Wnt11* ligands (Supplementary Fig. 9f) involved in either one or both canonical and non-canonical WNT pathways. Interestingly, most WNT signal senders were epithelial cells, and receivers were fibroblasts, while non-canonical *Wnt5a* and -*Wnt11* signals were primarily restricted to senders and receivers within fibroblasts. On the other hand, in the stomach, *Wnt4* and *Wnt5a* gene expression were observed, with senders and receivers being bi-directional between epithelial and fibroblast compartments (Supplementary Fig. 9f). Further, we spatially validated one of the key L-R interaction predictions where the *Pdgfa* ligand is primarily sent by Sq1-2 of the esophagus and tuft/endocrine cell types of the stomach targeting different fibroblasts (Fig. 7f). We confirmed the presence of *Pdgfa* sender cells (epithelia) and PDGFRA-expressing receiver cells (fibroblast) in the vicinity in both the esophagus and stomach, suggesting possible interaction (Fig. 7g, h). In line with this, a previous study showed that PDGFA expressing intestinal epithelium signals with PDGFRA expressing stromal cells for proper villi formation during gastrointestinal development[59]. Together, our findings deciphered the direction of the communication network and the role each cell type plays during different developmental stages in the process of GE-SCJ histogenesis.

## Discussion

The tissue microenvironment, including stromal and immune cells, is critical in regulating organ specification, histogenesis, and maintaining healthy homeostasis[14,60,61]. During tissue injury, the microenvironment reprograms to restore damaged tissue[62,63]. However, if the damage-inducing stimuli persist, the tissue might develop adaptive phenomena such as metaplasia to cope with the triggers[5,64]. The GE-SCJ shows increased susceptibility to Barrett's metaplasia (BE) development. BE adaptation is characterized by the replacement of stratified squamous mucosa of the esophagus at GE-SCJ with the columnar type of epithelium. Interestingly, BE is associated with the enrichment of pathogenic microbes and carcinogenesis[65].

This study provides a systematic temporal analysis of the histogenesis and regulatory interaction of healthy GE-SCJ from the late embryonic gestation stage of E15 to adult mice at single-cell resolution. The single-cell transcriptomic atlas revealed the diversity of cell types and delineated the evolution and differentiation process of epithelial cells and the tissue-resident fibroblast niche. First, we unraveled the evolution of the epithelial cells during development at GE-SCJ. We discovered that the KRT8 + /KRT7 + primitive cells of E15 that express unique transcriptional signatures (*Sox11, Igf2, H19, Cldn6, Vcan*, and *Bex1*) are precursors of both squamous and columnar epithelial lineages at GE-SCJ. These KRT8 + /KRT7+ precursor cells differentiate into P63- and P63+ cells, which eventually get segregated by E19 as distinct P63 + /KRT5 + /KRT8$^{low}$/KRT7$^{low}$ squamous and P63-/KRT5-/KRT8$^{high}$/KRT7$^{high}$ columnar cell types and will be maintained in the adult GE-SCJ. Gain of TF activities, including *Gata6, Foxa1/2*, and *Hnf4a*, specifies precursor cells to columnar lineage while *Trp63, Sox2, Klf5* specifies differentiation towards squamous epithelium during branching into stomach and esophagus at GE-SCJ. Further, our single-cell analysis showed that embryonic precursor cell-associated gene signatures were not found in the adult GE-SCJ cell population. Thus, we show the emergence of two distinct epithelial lineage-committed stem cells that regulate the regeneration of the squamous and columnar epithelia of the adult GE-SCJ mucosa.

Signal crosstalk between the epithelium and underlying mesenchyme directs the cellular differentiation and lineage specifications during embryogenesis[31,66]. Here, we found that many fibroblast cell states were unique to developmental stages, while some were shared. Further, we found regional differences in the esophagus and stomach fibroblast populations. Although certain fibroblast

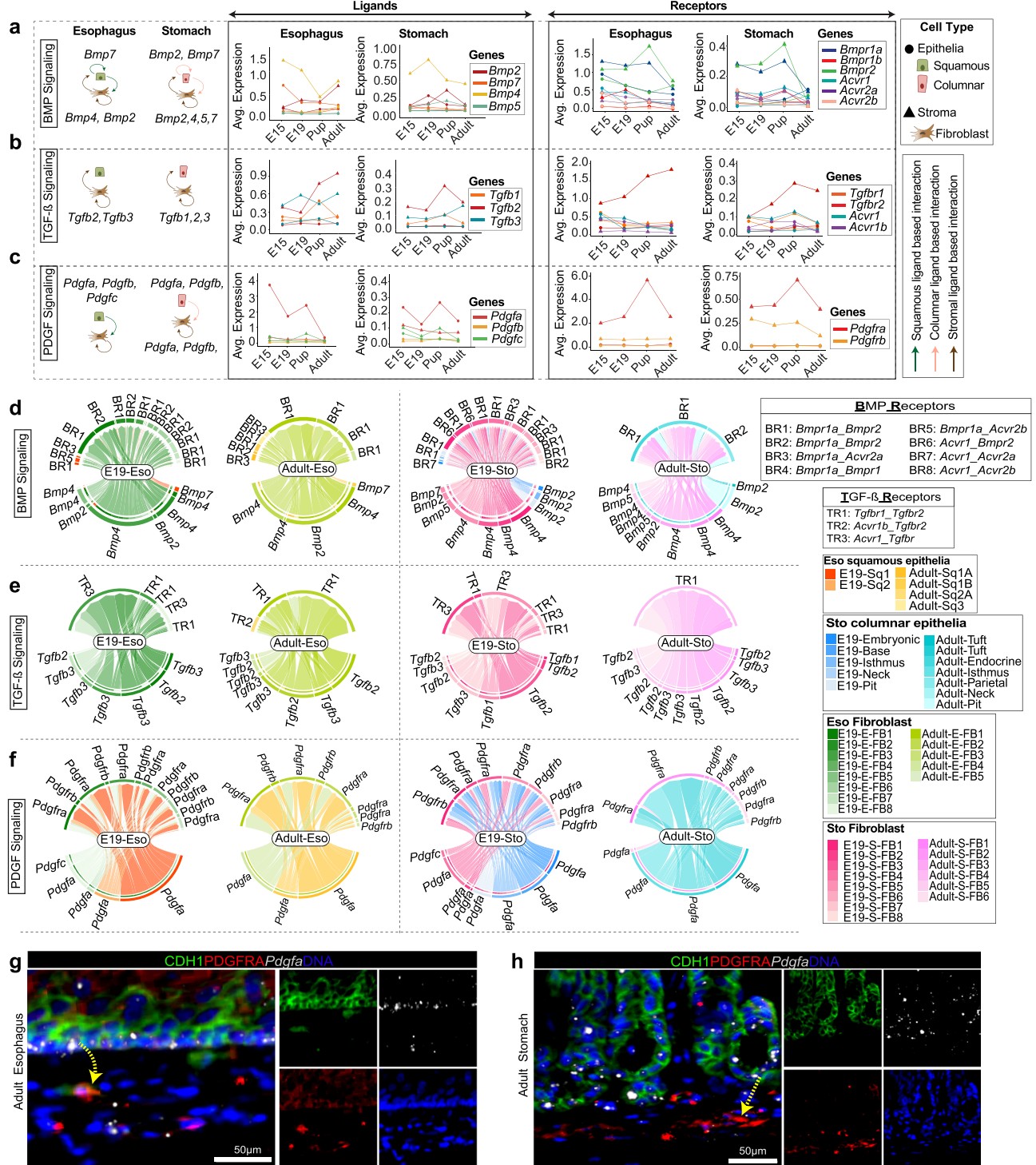

**Fig. 7 | Decoding spatiotemporal cell-cell interaction dynamics in pre- and postnatal gastroesophageal tissue. a–c** Graphical abstract of tissue-specific signaling directions between epithelia and fibroblasts (left); trend plots showing the mean expression dynamics of key ligands and receptors over time (right); for the following signaling pathways of interest: BMP (**a**) TGF-β (**b**) PDGF (**c**). Lines colored by gene with shapes representing the epithelial (circle) and fibroblast (triangle) cell population; arrows in graphical depictions show signaling direction and colored by signal origin: squamous epithelia (green), columnar epithelia (light pink) and fibroblast (brown). **d–f** Chord diagrams depicting inferred cell-cell communications mediated by multiple significant ligand-receptors between epithelia and fibroblast in esophagus and stomach at E19 and adult time points for BMP (**d**) TGF-β (**e**) PDGF (**f**) pathways; in lower half of the circos plot, outer bars colored by signal sending cell groups; inner bars colored by proportion of receiving cell groups; edges colored by signal senders. **g, h** Confocal images of the adult mouse esophagus (**g**) and stomach (**h**) tissue sections immunostained for CDH1 (green), PDGFRA (red), and smRNA-ISH probed for *Pdgfa* (white), and nuclei (blue). Images are representative of three biological replicates. Yellow arrow indicates the direction of predicted interaction between epithelial and fibroblast cells for PDGF signaling.

subpopulations share similar transcriptional signatures between the esophagus and stomach, their spatial location varies.

In particular, the abundance of the LRP6 receptor, involved in activating canonical WNT signaling, was higher in stomach epithelial stem cells than in esophageal basal stem cells. Strikingly, WNT inhibitor genes *Dkk2* and *Sfrp4* were found at higher levels in the esophageal fibroblast subpopulation. Further, basal cells of the esophageal squamous epithelium at GE-SCJ express DKK2 receptor *Kremen1*, which is required for the internalization of the DKK2-LRP6 complex, thus inhibiting cWNT signaling[67].

WNT signaling regulating morphogen, RSPO3 from myofibroblasts is known to regulate adult stomach epithelial stem cell regeneration[42]. We also found *Rspo3* expressing myofibroblasts underlying the columnar epithelium at the GE-SCJ. Interestingly, unlike columnar lined GE-SCJ mucosa where stem cells were proximal to *Rspo3* expressing myofibroblasts, the basal stem cells of the esophageal squamous epithelium and myofibroblasts were separated by wider lamina propria comprising fibroblasts expressing higher levels of *Dkk2*. Consistently, growth factors inducing WNT signaling inhibited the development and long-term maintenance of stratified squamous organoids from the esophagus while supporting the development and stemness of both human and mouse stomach organoids. Thus, spatially restricted differential expression of WNT signaling regulators underlying the epithelium is critical for adult GE-SCJ homeostasis.

Besides developmentally associated cWNT/beta-catenin pathways, we found ncWNT/Ca²⁺ signaling to be active in the columnar epithelium of GE-SCJ. However, extrinsic cWNT activators were found to be vital for the proliferation and regeneration of columnar epithelial stem cells of GE-SCJ. In contrast, the ncWNT/PCP pathway was predominantly active in the stratified squamous epithelium of the esophagus. Supporting this, we found a high-level expression of WNT ligands *Wnt4, Wnt7b*, and *Wnt10a* in esophagus epithelia, critical players in the ncWNT pathway[68]. Fibroblasts of both tissues were enriched for ncWNT ligand *Wnt5a*, suggesting its role in regulating fibroblast proliferation and apicobasal cell orientation in the gut region[69,70].

Further, the WNT/PCP signaling implicated in tissue morphogenesis and epithelial cell polarity during embryogenesis[71,72] was particularly active in the esophageal parabasal cells. However, it is not essential for esophageal stem cell regeneration and differentiation. Moreover, altered WNT signaling did not induce transdifferentiation between columnar and squamous epithelia. Thus, the distinct WNT signaling niche regulates the differential proliferation of these two epithelial lineages.

We also observed diverging gene expression patterns that differentially regulate and mediate epithelial-fibroblast signaling during GE-SCJ development, such as FGF, EGF, PDGF, and TGF-β. Fibroblasts predominantly regulate FGF and TGF-β signaling towards the epithelial cells. In contrast, PDGF and EGF signaling onset from epithelia to fibroblasts. Interestingly, higher expression of *Fgf10* in the fibroblasts underlying columnar epithelium and *Fgf7* in fibroblasts underlying the squamous epithelia suggested discrete roles of FGF ligands in both tissue types. Similarly, higher expression of receptors *Fgfr1* in both stomach and esophageal fibroblasts, while *Fgfr2* and *Fgfr3* in the embryonic and adult esophagus epithelia were observed, respectively. In addition, a recent study showed the GATA4-driven expression of *Fgf10* in stroma underlying columnar epithelium and SOX2-mediated expression of *Fgfr2* in the squamous epithelium in establishing squamous and columnar epithelium at GE-SCJ during development[32], conforming to our findings.

Hedgehog signaling is implicated in both esophagus and stomach epithelial morphogenesis[41,73,74]. We found that the Hedgehog ligand gene *Shh* was predominantly expressed in the early esophagus epithelial cells, indicating its role in esophagus development. However,

stomach epithelial cells expressed both *Shh* and *Ihh*. Accordingly, *Shh* and *Ihh* in the differentiation and proliferation of parietal, zymogenic, mucus neck, and pit cells of antrum stomach epithelium was suggested previously[75]. EGF ligands such as *Hbegf, Tgfa*, and *Btc* were expressed in the differentiated cells in both pre- and postnatal esophageal epithelia and interact with intermediate or parabasal epithelial cells of the esophagus and underlying fibroblast subpopulation. In the stomach, except neck-like cells, most E19 cells show interactions with receptors present in both epithelial and fibroblast cells. Further, the observed decreasing gradient in the expression of BMP pathway genes correlates with its regulatory roles in organ morphogenesis[40,75], however, its role in regulating GE-SCJ development needs further mechanistic evaluations. Extracellular matrix modulator, TGF-β signaling[76], is predominantly regulated by the fibroblasts underlying both columnar and squamous epithelium. However, its ligand expression in the esophageal stroma increased during the development, indicating stromal TGF-β signaling mediated epithelial regulation.

In conclusion, our study comprehensively delineates epithelial and fibroblast evolution and their interaction landscape during GE-SCJ histogenesis. In particular, this study emphasized the implications of stromal niches in evolving and controlling distinct squamocolumnar epithelial stem cells and their differential regeneration (Fig. 8). These insights pave the way to elucidate how non-resident epithelial type outgrows as a precancerous metaplasia at GE-SCJ and serve as invaluable resources for studies involving gastroesophageal disorders and early cancer events in other similar tissues.

## Methods

### Mice
All animal procedures were approved by the national legal and institutional authorities (Landesamt fur Gesundheit und Soziales (LaGaSo), Berlin, Germany, G 0026/17) at the Max Planck Institute for Infection Biology, Berlin, Germany. Wild-type C57BL/6 female mice were from Jackson Laboratory. *Krt5-CreERT2; Rosa26-tdTomato* and *Krt8-CreERT2; Rosa26-tdTomato* strains were generated as described in[14]. *Axin2CreERT2*[77] mice were bred to *Rosa-tdTomato* mice[78] to generate *Axin2-CreERT2; Rosa26-tdTomato* mice. Cre was induced by administering tamoxifen (Sigma, T5648) intraperitoneally (0.25 mg/g body weight in 50 µl corn oil) (Sigma, C8267) at week 4 for two consecutive days. Mice were euthanized at 14–20 weeks, and the gastroesophageal tissue was removed for further analysis. The stomach isolated from the postnatal mice or embryonic days 13, 16, and 19 were used for organoid culture or fixed with 4% PFA (Sigma, 441244) for 1 h at RT. Experiments were performed in at least three biological replicates per condition. The animals were housed in autoclaved micro-isolator cages, where they had access to sterile drinking water and chow ad libitum. Mice were bred within the animal care facility, maintaining a 12 h light/12 h dark cycle, and ensuring a controlled environment with a temperature of 22.5 ± 2.5 °C and humidity at 50 ± 5%.

### Cell line
3T3-J2 cells (mouse embryonic fibroblasts), generously provided by Craig Meyers, were cultured in HEPES-buffered Dulbecco's modified Eagle's medium (DMEM) (Gibco, 10938-025). The culture medium was supplemented with 10% fetal calf serum (FCS) (Biochrome, S0115), 2 mM glutamine (Gibco, 25030081), and 1 mM sodium pyruvate (Sigma, S8636). Cells were maintained at 37 °C in a humidified incubator with 5% $CO_2$.

### Organoid culture and maintenance
The Department of Hepatology and Gastroenterology, Charité University Medicine, Berlin, Germany, provided human esophagus, stomach, and Z-line (GE-SCJ) samples. Usage for scientific research was approved by their ethics committee (EA4/034/14); informed consent was obtained from all subjects.

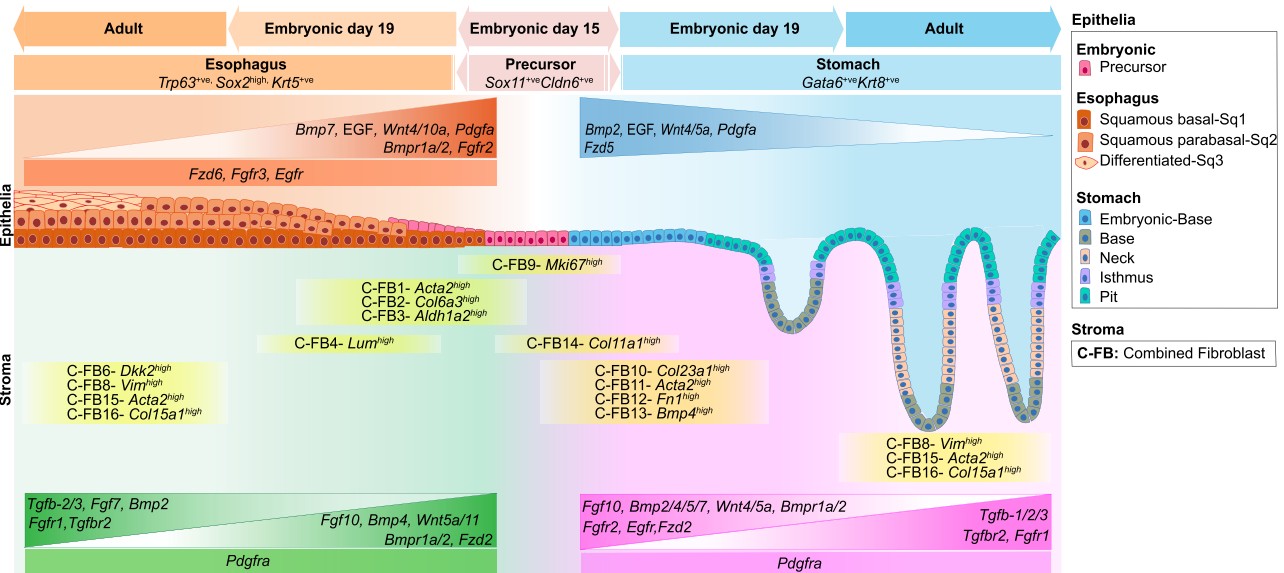

**Fig. 8 | Graphical summary of the spatio-temporal signaling events during gastroesophageal histogenesis.** The illustration provides a comprehensive overview of the critical signaling events occurring during the evolution of gastroesophageal epithelia and fibroblasts at the GE-SCJ. In this depiction, we elucidate the differentiation pathways of esophagus and stomach epithelial cells, starting from their precursor cells expressing *Sox11/Cldn6+* markers at embryonic stage 15. In the upper left portion of the panel (orange shades), precursor cells undergo differentiation into stratified squamous epithelial lineage cells (characterized by *Trp63+/Sox2+/Krt5+* markers) as they progress through development, culminating in the formation of parabasal and terminally differentiated cells in the adult phase. Conversely, the upper right panel (blue shades) illustrates the evolution of stomach columnar epithelial cells (marked by *Gata6 +/Krt8 +*). As development progresses, they organize into gland units with distinct epithelial regions such as the base, neck, and pit. In the lower panels, we observe the divergence in the distribution of esophageal (green shades) and stomach (magenta shades) stroma, featuring distinct subsets of fibroblasts with unique transcriptional signatures throughout development. Additionally, the regulation of key signaling pathways in the epithelia and fibroblasts, including WNT, BMP, EGF, FGF, PDGF, and TGF-ß pathways, with information on their signaling gradient established during gastroesophagus histogenesis are shown. C-FB indicates the combined fibroblast subclusters, as in Fig. 3b.

## Esophageal organoids

Human esophagus tissue or mice esophageal tissues (cut open along the esophagus tube) were washed three times with ice-cold sterile 20 ml PBS. Tissue was transferred to a 100 mm dish and minced with sterile scissors to small pieces in presence of 500 µl of pre-warmed 0.5 mg/ml collagenase type II (Calbiochem, 234155) solution. The tissues were then transferred to a 15 ml tube containing 5 ml of pre-warmed 0.5 mg/ml collagenase type II solution and incubated for 45 min at 37 °C, 180 rpm in a horizontal position in an orbital shaker incubator. Cells were centrifuged at 1000 g for 6 min at 4 °C. Supernatant was discarded, and cell pellet was resuspended with 5 ml of TrypLE Express solution (Gibco, 12605-028) and incubated for 15 min at 37 °C, 180 rpm in a horizontal position in an orbital shaker incubator. Cell aggregate was dissociated by pipetting up and down 20 times with a 1 ml pipette tip and 5 ml ice cold ADF + + medium (ADF medium (Gibco, 12634) supplemented with 12 mM HEPES (Gibco, 15630080), 1% GlutaMax (Gibco, 35050-038) was added. The cell suspension was passed through a 70 µm cell strainer (Corning, 352350), and the number of cells were counted.

For mice, 30,000 cells were mixed with 150 µl of Matrigel (Corning, 356231) in a ice cold Eppendorf tube, and placed 50 µl of mix as a one drop per well at the center of well in triplicates of pre-warmed 24 well plate and incubated for at least 15 min at 37 °C incubator. After Matrigel polymerization, added 500 µl per well mouse esophagus 3D medium: ADF + + containing 1% B27 (Gibco, 17504044), 1% N2 (Gibco,17502048), 50 ng/ml murine EGF (Invitrogen, PMG8041), 100 ng/ml murine noggin (Peprotech, 250-38-100), 100 ng/ml FGF-10 (Peprotech, 100-26-B), 1.25 mM N-acetyl-L-cysteine (Sigma, A9165-5G), 10 mM nicotinamide (Sigma, N0636), 2 µM TGF-β R kinase Inhibitor IV (Calbiochem, 616454), 10 µM ROCK inhibitor (Y-27632) (Sigma, Y0503), 10 µM Forskolin (Sigma, F6886) and 1% penicillin/streptomycin (Gibco, 15140-12), and incubated for one week in 5% $CO_2$ incubator

at 37 °C. Media was replaced with a new esophagus medium every 3 days. For passaging, Matrigel was removed by washing with ice cold ADF + +, and single cells were generated by incubating organoids with warm 500 µl of TrypLE for 20 min at 37 °C shaker and gentle disruption by pipetting up and down for 30 times using 200 µl tip. 5000 cells were seeded per 50 µl Matrigel per well of 24 well plate with esophagus medium.

For human, 50,000 cells per well were mixed with a 3 ml of human 3D-esophageal medium, having similar media component as above with minor modification, where EGF and noggin were replaced with human forms of 10 ng/ml EGF (Invitrogen, PHG0311) and 100 ng/ml noggin (Peprotech, 120-10 C). Cells grown on collagen type 1 (Sigma, C3867) treated 6 well plate until reaching 70% cell confluence by incubating for 12–14 days in a 5% $CO_2$ incubator at 37 °C. For passaging, cells were splitted into 1:2 ratio using 5 ml of TrypLE treatment and after wash, cells were seeded on Gamma irradiated 3T3-J2 cell line on a T25 flask. For organoid generation, 20,000 cells per 50 µl Matrigel per well in triplicates seeded and added with the human esophageal medium, incubated for 12–14 days. Patient-derived esophageal stem cells were enriched and expanded in 2D from 5 patients for four passages before aliquoting and biobanking. We could generate 3D organoids from these stem cells that were passaged 3-4 times in the esophagus medium.

## Stomach organoids

Corpus region of the mouse stomach was cut and washed thrice with ice cold PBS. Next corpus tissue was incubated with 0.5 mM DTT (Merck, 10197777001) /3 mM EDTA (Mecrk, E9884) in PBS solution (Gibco, 14190-169) for 90 min at RT. Corpus tissue was transferred to a 15 ml tube containing ice-cold PBS and stomach glands were isolated by shaking the tube vigorously for 1 min and number of glands were counted. 300 glands were transferred to the Eppendorf tube,

supernatant was discarded by centrifugation at 400 g for 6 min at 4 °C. Gland pellet was mixed with 150 μl of Matrigel and seeded 100 glands per 50 μl Matrigel per well in a triplicate of 24 well plate and incubated for 15 min at 37 °C. After polymerization of Matrigel, added 500 μl per well of mouse 3D-stomach media containing ADF + + medium supplemented with R-spondin1 (25%) and WNT3A (25%)-conditioned medium, 1% B27, 1% N2, 50 ng/ml murine EGF, 100 ng/ml murine noggin, 100 ng/ml FGF-10, 1.25 mM N-acetyl-L-cysteine, 10 mM nicotinamide, 2 μM TGF-β-R kinase Inhibitor IV, 10 μM ROCK inhibitor (Y-27632), 10 mM gastrin (Sigma, G9145) and 1% penicillin/streptomycin and incubated for 1 week. For every 3 days, the media was replaced with fresh mouse 3D-stomach media. For organoid passaging, Matrigel was removed by resuspending with 5 ml of ice cold ADF + +, and centrifuged at 400 g for 6 min at 4 °C. Organoid pellet was added with 500 μl of TrypLE and incubated for 10 min at 37 °C in shaker and organoids were dissociated by pipetting up and down for 5 times using 200 μl tip. 3 ml of ADF + + was added, split at a ratio of 1:4 and transferred to Eppendorf tube, seeded in 50 μl Matrigel per well of 24 well plate with 3D-stomach medium.

For human stomach organoid culture, stomach gland cells were isolated similar to esophagus tissue as above with minor modification by mincing tissue, 0.5 mg/ml collagenase type II solution treatment for 30 min followed by TrypLE treatment for 15 min. Polymerized Matrigel was overlaid with 500 μl of 3D-stomach medium similar to mouse stomach medium but containing human forms of 10 ng/ml EGF, 100 ng/ml noggin, incubated for 1 week, and the media was changed every 3 days. Human stomach organoids were passaged similar to mouse stomach organoids as above.

## Organoid-forming efficiency and size analysis

Epithelial cells were counted, and 5000 cells were resuspended in 50 μl of Matrigel in triplicates to generate organoids as described above. One week after plating, images were acquired from the whole well, and the number and diameter of formed organoids were determined using ImageJ to calculate the organoid-forming efficiency and measurement of size.

## Immunofluorescence, smRNA-ISH, and microscopy

Tissues and organoids were fixed, paraffinized, and immunostained or used for smRNA-ISH labeling as described previously by us[14]. Images were acquired with AxioScan.Z1 tissue imager (Zeiss), Keyence BX800, or a confocal microscope (Leica), processed with Adobe Photoshop or Zen 2.3, and analyzed using Image J. The antibodies and dilutions are listed in Supplementary Data 1.

## Microarray expression profiling

RNA was isolated from organoids resuspended in Trizol (Invitrogen, 15596026) according to the manufacturer's protocol. Microarrays were performed as single-color hybridizations on Agilent-028005 SurePrint G3 Mouse GE 8x60K, and probe intensities were obtained using Agilent Feature Extraction software. Raw data were background corrected, quantile normalized, and differential gene expression (p-Value < 0.05 and 1.5-fold change) was analyzed using the R package LIMMA. Over-representation analysis was performed using compare Cluster function in ClusterProfiler with default setting as significance cutoff and an adjusted p-value < 0.05.

## Single-cell isolation, scRNA-seq library preparation, and MULTIseq

Following gut extraction from pre- (E15, E19) and postnatal (Pup, Adult) mice, the esophagus, gastroesophageal squamocolumnar junction, and stomach were cut out. Tissue samples were washed in sterile PBS and minced with scissors. Minced tissue was processed separately by incubating in 0.5 mg ml$^{-1}$ collagenase II in a shaker (45 min, 37 °C).

Tissue and dissociated cells were pelleted (7 min, 1,000 g, 4 °C), and the supernatant was discarded.

For single-cell sequencing, we combined cells isolated from organoids or tissues derived from three mice and processed each tissue separately. Cells derived from in vivo tissue or organoids were resuspended in TrypLE Express and incubated in a shaker (15 min, 37 °C). The pellet was resuspended in Advanced DMEM/F-12 (ADF) medium / 0.04% BSA-PBS solution and passed through a 40 μm cell strainer (BD Falcon, 352340).

Single cells from organoids were washed with 0.1% BSA in 1XPBS and used for cell multiplexing according to the MULTI-seq protocol[79]. Sample multiplexing and library preparation for tissue were performed using 10 x Genomics 3' CellPlex Kit and Single-Cell 3' v3.1 RNA-seq kit. Sequencing was performed in paired-end mode with an S1 100-cycle kit using a Novaseq 6000 sequencer (Illumina).

## Bioinformatic analysis of scRNA seq data derived from in vivo tissues

**Deconvolution of raw sequencing data and downstream analysis.** Using the CellRanger (v.6.1.1) software suit provided by 10X Genomics, raw sequencing data of samples were demultiplexed and processed using the standard 'cellranger multi' pipeline with default parameters. Transcripts were aligned using the 10X reference mouse genome build mm10-2020-A. The raw gene expression matrices obtained per sample were then coupled with R package Seurat (v.4.1.1) for downstream analysis[80]. Contrary to other samples, no cells of the adult stomach were assigned with CMO, potentially indicating poor cell handling, library quality, or sequencing quality, and excluded from the analysis. Next, we scrutinized for potential doublets by neglecting barcodes with <100 genes, >8500 genes, and >80,000 UMI counts. Low-quality cells with >20% of the UMIs derived from the mitochondrial genome were excluded. Additionally, two subgroups of cells that passed the initial filter with poor read counts were identified and excluded from the rest of the study, leaving 7,808 cells for further analysis.

The count data were normalized using a negative binomial regression model provided by the R package sctransform (v.0.3.3)[81]. In addition, the mitochondrial mapping percentage and cell cycle scores (calculated using the 'CellCycleScoring' function) were regressed out during data normalization and scaling. Dimensionality reduction and data clustering were performed using the 'RunPCA', FindNeighbors, and 'FindClusters' functions, which were then visualized by implementing a nonlinear dimensionality reduction via the 'RunUMAP' function.

From 7,808 cells, we identified 6 major cell types in the first round of clustering: stromal, squamous, columnar epithelia, immune, endothelial, and neural cells. Before analyzing epithelial and fibroblast cell populations to decipher their cellular subtypes, we integrated other esophagus and stomach epithelial and stromal data (see next section 2.) to compensate for the data gap in postnatal samples. We employed the Seurat SCT integration workflow using built-in functions to do this. The reciprocal PCA approach was then used to find integration anchors between the samples using 'PrepSCTIntegration' and 'FindIntegrationAnchors' functions. These determined anchors were utilized by the 'IntegrateData' function, which results in the integration of cells from multiple datasets by generating a batch-corrected gene expression matrix. Nevertheless, previous original read counts (raw and normalized) were stashed for downstream analysis like differential expression testing. 'RunPCA', 'FindNeighbors', and 'FindClusters' functions were used for dimensionality reduction and clustering, then visualized using the UMAP algorithm. As a result, for both esophagus and stomach samples at four different time points, 6,536 integrated cells were obtained, among which 3,879 cells were from the stroma and 2,657 cells were from epithelia. Finally, we re-clustered stromal and epithelial cell groups separately to identify the subpopulations within them by repeating the same workflow.

**Joint analysis of external scRNA-Seq data.** To add reinforcement to the gastroesophageal data analyses and validations, especially for the unaccounted postnatal stages, the following external scRNA-seq datasets were downloaded from a) GEO (Gene Expression Omnibus) database using accession codes: GSE116514 and GSE157694 for adult stomach stroma[82] and epithelia respectively[6]. b) From ArrayExpress repository with accession code: E-MTAB-8662 for postnatal esophagus epithelia[9]; c) From GSA (Genome Sequence Archive in BIG Data Center, Beijing Institute of Genomics, Chinese Academy of Sciences, http://gsa.big.ac.cn) with accession code CRA002118 for adult esophagus stroma[83]. Next, we normalized each dataset separately using the 'sctransform' approach by repeating the standard Seurat workflow. In addition, to remove any bias towards cell count distribution between datasets, we implemented a uniform approach where we down-sampled 10% of cells from each cluster within a dataset for further analysis. This ensures that cell-type information within each dataset is preserved. In contrast to the above methods, for the adult stomach epithelial dataset, for merging replicates, we used the standard Seurat integration workflow described in the previous section. Here, 50% of cells were downsampled from each cluster and utilized for further downstream analysis.

**Differential expression analysis and gene set enrichment analysis (GSEA).** To identify differentially expressed genes (DEG) between cell types/clusters, we used the 'FindAllMarkers' function from the R package Seurat using default settings. Later, the average expression of identified DEG was calculated across cell types/clusters and visualized using a heatmap. GSEA was performed on scRNA-seq count data using the default functions provided by the R package escape (v.1.4.1)[84].

**Cell-cell interaction analysis.** We used CellChat's (v.1.1.3) standard pipeline to systematically infer cell-cell communication between epithelial and stromal subsets of the esophagus and stomach across different time points. First, to identify potential interactions, we preprocessed the expression matrix using the in-built functions 'identifyOverExpressedGenes', 'identifyOverExpressedInteractions', and 'projectData' with default parameters. Next, we used the functions 'computeCommunProb', 'computeCommunProbPathway', and 'aggregateNet' to infer the communication network and calculate communication probabilities. In addition, to remove the effect of cell proportion during probability calculation, we set the parameter population size as true.

**Pseudotime analysis/cell transition trajectory and diffusion map analysis.** Differentiation trajectories in our data were reconstructed using the R package URD (v.1.1.1)[85]. The diffusion map was generated by calculating the transition probabilities between cells in the data using the R package destiny (v. 3.9.1)[86]. Several simulations were run to calculate pseudotime by defining a set of cells as the starting point (root) and possible endpoint (tip) based on expression levels of stem cell and highly differentiated markers, respectively, using the standard in-built functions. Finally, the developmental trajectories were built and visualized using the 'buildTree' and 'plotTree' functions.

**Transcription factor (TF) activity Inference.** To study the TF activities of epithelial stem cells between precursor and stem cell compartment across time points, we used the R package Dorothea with default settings(v. 1.4.2)[27].

**Organoids scRNA-seq data processing and analysis.** Sequencing data of mouse organoids were processed using the CellRanger (v3.1.0) pipeline using the commands "cellranger mkfastq" and "cellranger count" with default parameters to perform alignment against the mouse build mm10-2020-A genome assembly, UMI counting and for generating the feature barcode matrix. To demultiplex, i.e., to determine the sample origin of each cellular barcode, we used the R package deMULTIplex(v1.0.2) with default functions (https://github.com/chris-mcginnis-ucsf/MULTI-seq). As a result, each cell's origin was obtained and utilized for further downstream analysis. Next, using R package Seurat (v.4.0.0)[80], we split each unique sample into a separate Seurat object based on the MULTI-seq sample barcodes, which contained 765 cells from the esophagus and 90 cells from stomach samples designated for further downstream analyses. Normalization, dimensionality reduction, and clustering were done using the sctransform package with default functions. In addition, we identified a set of cells with erroneously annotated sample barcodes, which might be due to the negative cell reclassification during the demultiplexing process. Hence, we carefully assessed for the presence of such other cells (e.g., mix-up cells/doublets with substantial and coherent expression profiles of a hybrid transcriptome based on columnar and squamous epithelial marker gene expression (*Krt8/18* and *Krt5/14/6a/13*, respectively) and excluded them from further analysis. As an outcome, UMAP was derived from analyzing a total of 612 cells from the esophagus and stomach samples combined (Fig. 5r). Next, to model developmental trajectories, we used the standard pipeline provided by the R package Slingshot (v.1.6.1)[49] with default parameters.

### Statistics and reproducibility
GraphPad Prism (v.8) was used for statistical calculations and the generation of plots. The data are displayed as mean ± s.e.m. $p < 0.05$ was considered statistically significant.

### Reporting summary
Further information on research design is available in the Nature Portfolio Reporting Summary linked to this article.

## Data availability
Microarray and scRNA-seq data supporting this study's findings have been deposited in the GEO under accession codes GSE181409, GSE181411, and GSE227412, respectively. Previously published scRNA-seq datasets that were re-analysed here are available under accession codes GSE116514, GSE157694, E-MTAB-8662, and CRA002118. Quantitative data supporting this study's findings are available within the paper and its supplementary information. Source data underlying the graphical representations in Figs. 4e, f, h, i, 5e, f, i, j, Supplementary. Fig. 6f, g are provided in the Source Data file. Source data are provided with this paper.

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

## Acknowledgements

We thank M. Drabkina and K. Hoffman for technical assistance; I. Wagner for the microarrays; D. Son for help with sample preparations. We thank Core Unit Systems Medicine (SysMed), Uni Wuerzburg for scRNA-seq. N.K. is supported by Deutscher Akademischer Austauschdienst (DAAD), C.C. research is funded by DFG CH2527/2-1 and The Novo Nordisk Foundation (NNF220C0077183). Parts of the project were supported by the German Research Foundation in the CRC 1583 "DECIDE" in projects A08 and Z02, as well as in the RTG 2157 "3D Infect" in project 12N. The funders had no influence on the study design or analysis of the data.

## Author contributions

C.C., R.K.G. conceived and jointly supervised the study; N.K. R.K.G. and C.C. designed the experiments and performed and analyzed the data; S.M.K. and G.J. contributed to IHC experiments; N.K. and S.M.K. performed the single-cell preparation from tissue and organoids, sample multiplexing for scRNA seq; T.K., C.T., and A.-E.S. performed the scRNA-seq of organoids; P.G.P performed all the scRNA-seq bioinformatic analysis, and C.W. performed microarray analysis with the help of N.K., R.K.G and C.C.; V.B. contributed Axioscan imaging; H.-J.M. contributed microarray studies; T.F.M. provided the infrastructure and advice; C.J., M.B. and B.W. provided human samples; N.K. P.G.P., R.K.G., and C.C. wrote the manuscript.

## Funding

## Competing interests

The authors declare no competing interest.

## Additional information

[1]Laboratory of Infections, Carcinogenesis and Regeneration, Medical Biotechnology Section, Department of Biological and Chemical Engineering, Aarhus University, Aarhus, Denmark. [2]Department of Microbiology, University of Würzburg, Würzburg, Germany. [3]Department of Molecular Biology, Max Planck Institute for Infection Biology, Berlin, Germany. [4]Helmholtz Institute for RNA-based Infection Research (HIRI), Helmholtz-Center for Infection Research (HZI), Würzburg, Germany. [5]University of Würzburg, Faculty of Medicine, Institute of Molecular Infection Biology (IMIB), Würzburg, Germany. [6]Surgical Clinic Campus Charité Mitte, Charité University Medicine, Berlin, Germany. [7]Department of Hepatology and Gastroenterology, Charité University Medicine, Berlin, Germany. [8]These authors contributed equally: Naveen Kumar, Pon Ganish Prakash. ✉e-mail: cindrilla.chumduri@bce.au.dk

