## [Peer Review File · Nature Communications]

Decoding Spatiotemporal Transcriptional Dynamics and Epithelial Fibroblast Crosstalk during Gastroesophageal Junction Development through Single Cell AnalysisREVIEWER COMMENTS

Reviewer #1 (Remarks to the Author):

The authors dissected the developmental stages of the squamous columnar junction between the esophagus and the stomach at single-cell resolution in mice. They identified a precursor population which gives rise to gastric and esophageal epithelium during embryogenesis and is lost in adult mice. Additionally, they analyzed the underlying fibroblast population which are distinct between the stomach and esophagus. Lastly, they provide a resource of ligand receptor interaction between fibroblasts and epithelial cells.

A better characterization of the cellular composition of the gastroesophageal junction during development is of great interest for the community.

Major criticisms:

- The use of the terminology “gastroesophageal junction (GEJ)” in mice does not feel justified, as the transitional epithelium is not located between the esophagus and the stomach as also highlighted by the authors. A better terminology would be squamous columnar junction as others have used previously (PMID: 29019984).

- It would be nice to validate the expression of some precursor marker genes by histology staining. That would strengthen the conclusion.

- Some conclusions are very speculative. I would recommend tuning those down:

- page 6:

“Corroborating to scRNA seq data in Fig. 1, we confirmed that adult GEJ comprises distinct squamous and columnar epithelial lineages with independent regenerative capacities.”

The regenerative process was not shown or proven by the scRNA seq data. The scRNA seq data measured a snapshot at the given time.

- page 8:

“Interestingly, TF Nanog, ... suggesting they may play a role in lineage commitment to the squamous epithelium.”

Can authors speculate more about their potential role? In particular, since some members are reported to be important for well-known cell types. Pax5 is e.g. a lineage commitment factor for B cells. It would be good to validate these data with some additional staining. This would strengthen the conclusion.

- page 14:

“The ligands of FGF show differential expression in fibroblasts where expression of Fgf7 found in the esophagus and Fgf10 in the stomach, indicating it may play a role in regulating GEJ.”

This is a speculation and was not proven. If the presence of Fgf7 and Fgf10 guide the development into the gastric or esophageal tissue at the squamous columnar junction during development cannot be deduced from the current data.

- page 18:

“a decreasing gradient of BMP pathway ... indicate its role in GEJ development”. That is a speculation as it correlates with expression changes but was not mechanistically shown.

- The assembly of the figures seems a bit rushed as quite some errors seem to be present. Please see my comments below. These require some work but are largely easily fixable.

- I am aware that the authors used some existing algorithms such as cell chat to analyze their data.

However, the data representation is not clear. This concerns mainly Fig. 5b and c, and entire Fig. 6.

- Figs. 5b and c:

It is not clear what is incoming or receiving? What does relative strength refer to? As far as I understand if e.g. Notch is brown in adult epithelium and absent in fibroblast, it means it is expressed only in the epithelium. If it is labelled brown in E19 epithelium and fibroblast, it is produced in both tissue types.

How do I deduce now receiver and targeted cells based on these plots? In any case, common genes that are present in b and c should be located on the same height for the ease of the reader.

- Fig. 6a-g:

This figure is simpler as it shows the interpretation of the data obtained by the cell chat algorithm in a cartoon on the left and the expression data on the right. It is, however, not clear what exactly is plotted on the y-axis as it is not stated. Moreover, error bars are missing. Since the statement is that the ligands act on the same cells, it seems justified to make the same graphs for the corresponding receptors next to the ligand.

- Fig. 6h-l:

While it may represent a good resource for the community, these figures are not interpretable. There are too many color codes. It is impossible identify the cell types of interest. These graphs need to be redone in simpler versions.

Minor criticisms:

- In the introduction the authors mentioned some literature references for scRNA seq data but left others out such as PMID: 34795 for Barrett's esophagus or PMID: 36717627 for gastric tissue.

- Fig. 1d: For the embryonic cell cluster authors have calculated cell distribution from the different time points (Fig. 1r). Could they generate a similar plot also for the other clusters in the supplement. That could be informative to see how many specialized cells are already present at what developmental stages.

- Fig 2k:

Could authors clarify what happens to cells on the right side, which seem to get lost after embryonic stage E19. Are these E19 specific?

- Concerning organoid data:

- Mouse esophageal cultures were already previously published (PMID: 25373907). Please cite them.

- There is also already a publication about healthy human esophageal organoids (PMID: 34795059).

Could you specify how your culture medium differs from them?

- Establishing long term human esophageal cultures are of great interest as this is still challenging. Could authors provide data on how many passages human esophageal organoids could be maintained and how many different patients they have tried?

- Page 11:

The authors mentioned they detected 34,393 genes. Generally, people refer with genes to protein coding genes. I assume they also included long non-coding RNAs etc. It would be great to specify that or restrict the analysis on protein-coding genes.

- Gene set enrichment analysis in Fig. 5a:

Some terminologies are highlighted but others are not. Can author explain what the others are and why

only the selected ones were picked? In particular, since some entries have a similar expression pattern to the picked ones such as the two above 6.

- The authors mentioned couple of times the “panoramic view” of their data. I am not sure what they are referring to with this. The terminology does not seem very scientific.
- Some findings could be better put in context with the literature. E.g. Fgf7 is important for esophageal epithelium and Fgf10 for gastric one. This is also reflected in published organoid culture medium (gastric organoids need Fgf10: PMID: 25307862, PMID: 25539675, esophageal organoids grow with Fgf7: PMID: 34795059).

Comments to figure representation and suggestions:

- The color choices for the representation of scRNA seq data is often poorly chosen as it makes it difficult to follow their conclusion (e.g. Fig. 1c, 1l, 2i-l, 3a, S3a etc). It would be better stick to a color for either the same tissue and use different shades for the different time points. Alternatively use the same color for time points with different shades for the respective tissues.
- Fig. 1h: scale of y-axis is missing.
- Fig. 1i, j, and k: Krt7 is highlighted in red in Fig. 1i and 1k but white in Fig. 1j whereas Krt8 is highlighted in red there. Please stick to the same color code which makes it easier for the reader.
- Concerning all histological staining: please stay consistent with labeling. For mouse tissue use mouse nomenclature (Krt7) and not KRT7 whereas human sections should be labeled with the human nomenclature (KRT7).
- Fig. 1i-k and 2a-c:
 - The left panels are too small, and it is difficult to see anything in them. Please provide bigger versions of them.
 - For the ease of the reader, it would be great to label forestomach, columnar epithelium and esophageal remnants in the pictures.
- Fig. 2l: The graphics is confusing. Some categories come up more often. Please replace it with something more understandable.
- The legend of heatmaps is confusing:
 - Fig. 2m, 4t-v, 5d, S1b,d, S2c-d : the legend is “Avg. Expr “
 - Fig. 2n: the legend is “TF Activity Score”
 - Fig. 3e: the legend is “Expression”
- ◇ I strongly assume that in each of these cases, the Z-score that is displayed. Please clarify this.
- Fig. 3d, e: In Fig. 3d authors show that fibroblast clusters can be grouped into four subtypes but in Fig. 3e each cluster has a unique expression pattern. Could the authors clarify that and eventually provide a heatmap where the similarity can be seen. Moreover, it would be good to group the clusters in Fig. 3e according to the major subtypes in Fig. 3d.
- Fig. 3f and g are identical except for the highlighting of tissue type.
- Fig. 3j: Is it really average exp value or percentage that is displayed? Anyway, could you please include error bars?
- Fig. 5d:

The authors used a curated list for signaling pathways but unfortunately they did not group receptors or ligands of the same class next to each other, which seems to make more sense. E.g. Fzd receptors or Fgfrs are all over the place as well as Notch ligands.

Reviewer #2 (Remarks to the Author):

Throughout organogenesis, epithelial and the surrounding mesenchymal tissues highly communicate each other to induce complex and interconnected organs. Neighboring tissues secrete various paracrine factors to direct tissue growth and maturation of the organs. The gastroesophageal junction (GEJ) is defined as the junctional structure between esophagus and stomach, which can be a hotspot of Barrett's esophagus. Therefore, the understanding of the process of GEJ development is important to clarify the mechanisms of metaplasia and cancer development. However, as described in manuscript, the process of GEJ development has not fully understood yet.

In this study, Kumar et al conducted single cell RNA sequence (scRNA-seq) analyses to investigate cell heterogeneity and signaling network governing the development of gastroesophageal junction (GEJ). The analyses of epithelial and mesenchymal lineage revealed that there is a diversity in gastroesophageal cells during the development. They further showed different Wnt dependency between esophageal and gastric epithelium by taking advantage of both mouse model and organoid technologies. Finally, the study has provided signaling interplay between epithelium and fibroblasts at GEJ. Overall, the results are informative and interesting for the readers, especially in the field of organogenesis, of nature communications. However, the data should be more spatially validated with staining of respective markers. The authors are also encouraged to address several concerns as written below.

- scRNAseq analyses revealed the diversity of cell types during GEJ development (Sqs, FBs). However, the distribution of these subtypes is unclear. The authors should perform immunostaining or in situ hybridization for the respective markers to provide spatial information of these cell types.

- Based on scRNA seq and cell-cell communication analyses, the authors suggest ligand-receptor interaction between epithelial cell types and fibroblast cell types during GEJ development. However, these analyses do not demonstrate that sender cells and receiver cells are located closely enough to signal each other. It would be better to stain the ligand-receptor pairs, and target genes of signaling pathways.

- The study shows that Wnt signaling is activated in esophageal compartment but inactivated in stomach. Fig. S6 shows ligand-receptor pairs of Wnt signaling, but authors did not mention these in result sections. Authors should describe what Wnt ligands, receptors and antagonists in addition to Rspo3 and Dkk2 are expressing at GEJ.

- Fig. 3k, n: Cell types are not clear. Marker genes (e.g. SMA, TAGLN) should be co-stained to show the cell types which express Rspo3 and Dkk2.

- Fig. 3s: It is difficult for the readers to see which tissue is activated in Wnt signaling. The authors should perform double or triple co-staining for Axin2-lineage with epithelial and/or mesenchymal e markers.

- Fig. 6: The data of signaling networks are too complex at first glance. In order to summarize critical points, table and/or cartoons would be helpful for the readers.

REVIEWER COMMENTS

Reviewer #1 (Remarks to the Author):

The authors dissected the developmental stages of the squamous columnar junction between the esophagus and the stomach at single-cell resolution in mice. They identified a precursor population which gives rise to gastric and esophageal epithelium during embryogenesis and is lost in adult mice. Additionally, they analyzed the underlying fibroblast population which are distinct between the stomach and esophagus. Lastly, they provide a resource of ligand receptor interaction between fibroblasts and epithelial cells.

A better characterization of the cellular composition of the gastroesophageal junction during development is of great interest for the community.

Response: Thank you very much for your support and thorough feedback. We appreciate your assessment of the novelty and significance of our work.

Major criticisms:

- The use of the terminology “gastroesophageal junction (GEJ)” in mice does not feel justified, as the transitional epithelium is not located between the esophagus and the stomach as also highlighted by the authors. A better terminology would be squamous columnar junction as others have used previously (PMID: 29019984).

Response: We appreciate and agree with the reviewer’s suggestion, regarding the terminology. We have revised the term gastroesophageal junction (GEJ) to gastroesophageal squamocolumnar junction (GE-SCJ) where appropriate throughout the manuscript to more accurately reflect the anatomical and cellular characteristics of the region under study.

- It would be nice to validate the expression of some precursor marker genes by histology staining. That would strengthen the conclusion.

Response: We have now validated the expression of the precursor marker genes *Sox11* and *Cldn6*, as initially identified in figures 1f, h. These genes were marked with specific smRNA-ISH probes and co-immunostained with the squamous epithelium-specific marker KRT5 and the columnar epithelium-specific marker KRT8 in both embryonic and adult GE-SCJ tissues. This newly generated data confirms our conclusions and is presented in the revised manuscript (Fig. 1i, Supplementary Fig.2a-b).

- Some conclusions are very speculative. I would recommend tuning those down:
• page 6:

“Corroborating to scRNA seq data in Fig. 1, we confirmed that adult GEJ comprises distinct squamous and columnar epithelial lineages with independent regenerative capacities.” The regenerative process was not shown or proven by the scRNA seq data. The scRNA seq data measured a snapshot at the given time.

Response: We have revised the sentence to better reflect the data presented as “*Corroborating the scRNA seq data in figure 1, we observed that the adult GE-SCJ comprises two distinct squamous and columnar epithelial lineages, each characterized with lineage-specific gene expression pattern.*” Page, Lines:18-189 (in track change mode)

• page 8:

“Interestingly, TF Nanog, ... suggesting they may play a role in lineage commitment to the squamous epithelium.”

Can authors speculate more about their potential role? In particular, since some members are reported to be important for well-known cell types. Pax5 is e.g. a lineage commitment factor for B cells. It would be good to validate these data with some additional staining. This would strengthen the conclusion.

Response: Thank you for your suggestion.

Regarding *Pax5 as an example*, our data indicates its enrichment in specific cell states within both squamous and columnar lineages. Although the role of PAX5 in epithelial regulation remains unclear, its known interactions with E-cadherin (PMID: 28076843) and its implication in epithelial-to-mesenchymal transition suggest it may influence cell patterning and regenerative processes. However, investigating the precise role of PAX5 and other TFs such as *Nanog*, *Tead1*, *Prdm14* in lineage commitment within the squamous and columnar epithelia is beyond the current scope of our study and presents a substantial avenue for future research.

We have modified the text and further validated the transcription factors SOX2 and GATA6 that show distinct activity in squamous and columnar epithelial lineage respectively (Fig. 2m, Supplementary Fig.4j). SOX2 expression was confirmed to be high in the squamous epithelium, aligning with previous findings (PMID: 33495473), and GATA6 was found to be highly expressed in the columnar lineage at the GE-SCJ. We found GATA6 expression specifically in the lower part of the stomach gland, suggesting that it might play a role in columnar stem cell maintenance and differentiation that needs to be further elucidated experimentally. In line with this observation, other studies have also shown that GATA6 regulates intestinal epithelial proliferation, lineage maturation (PMID: 22733991, PMID: 24929016), and BMP repression (PMID: 24952462).

The text is modified in the results section of the revised manuscript as “*SOX2 expression was confirmed to be high in the squamous epithelium, aligning with previous findings³¹, and GATA6 was highly expressed in the columnar lineage at the GE-SCJ (Fig. 2m, Supplementary Fig. S4j). GATA6 expression was confined specifically to the lower part of the stomach gland, suggesting that it might play a role in columnar stem cell maintenance and differentiation that needs to be further elucidated. In line with this, other studies have shown that GATA6 regulates intestinal epithelial proliferation, lineage maturation, and BMP repression³²⁻³⁴. Further, TFs such as Nanog, Tead1, Prdm14, Pax5^{35,36} activity were enriched in the early-stage squamous epithelium and specific cell states of columnar epithelia (Fig. 2l, and Supplementary Table 4). However, their mechanistic role in lineage commitment within the squamous and columnar epithelia is unclear and an avenue for future research.*”. Page 9, Lines 255-266 (in track change mode)

• page 14:

“The ligands of FGF show differential expression in fibroblasts where expression of Fgf7 found in the esophagus and Fgf10 in the stomach, indicating it may play a role in regulating GEJ.”

This is a speculation and was not proven. If the presence of Fgf7 and Fgf10 guide the

development into the gastric or esophageal tissue at the squamous columnar junction during development cannot be deduced from the current data.

Response: We acknowledge that our initial discussion may have suggested a conclusive role of FGF ligands in guiding tissue specificity at GE-SCJ.

Although *Fgf7* is predominantly found in the esophagus and *Fgf10* in the stomach, which suggests a possible role in developmental guidance at the GE-SCJ, such a role cannot be conclusively established based on our findings. It is important to note that FGF7 is critical for the development of esophageal epithelium and FGF10 is vital for gastric epithelium—reflected by their inclusion in the respective organoid culture media (PMID: 34795059, PMID: 25307862, PMID: 25539675).

Further, previous studies have shown that higher expression of *Fgf10* has been observed in stomach fibroblast cells. This observation aligns with findings from another study (PMID: 33495473) that showed culturing embryonic junctional cells with FGF10 favors the formation of columnar organoids, a characteristic more consistent with gastric tissue, as also indicated by the use of FGF10 in adult stomach organoids culture medium (PMID: 25307862, PMID: 25539675). Conversely, embryonic esophageal epithelial cells treated with FGF10 demonstrated a diminished differentiation in esophageal organoids.

Furthermore, the esophagus epithelial cells show a higher expression of *Fgfr2*, a receptor that binds to FGF7, suggesting a regulatory mechanism for squamous epithelial cell proliferation and differentiation, which is corroborated by the use of FGF7 in esophageal organoid culture media (PMID: 33691112, PMID: 34795059). While our data are suggestive, they should not be interpreted as definitive evidence of FGF7 and FGF10's involvement in developmental guidance at the SCJ, but rather as a potential area for further investigation.

We have amended the text in the results section as “*The distinct expression profiles of FGF ligands in fibroblasts, with Fgf7 highly expressed in the esophagus and Fgf10 in the stomach, suggest a regulatory role in the GE-SCJ. The expression patterns of Fgf7 and Fgf10 align with their requirement for esophageal¹³ and stomach^{47,53} epithelium, as evidenced by organoid studies^{6,31}—nonetheless, their precise contribution to GE-SCJ development remains to be elucidated.*” page 17, Line 506-512

• page 18:

“a decreasing gradient of BMP pathway ... indicate its role in GEJ development”. That is a speculation as it correlates with expression changes but was not mechanistically shown.

Response: We have revised our discussion regarding the BMP pathway. Previous studies have elucidated the multifaceted roles of BMP signaling in gastrointestinal development. For instance, BMP signaling has been implicated in the differentiation process towards parietal cells in the stomach and is associated with the proliferation of neck cells in the adult stomach epithelium (PMID: 20826155). In the esophagus, BMP signaling is critical for the differentiation of basal to suprabasal cells (PMID: 21068065). Interestingly, during embryonic development, an increase in BMP signaling is reported to inhibit the transition from columnar to stratified squamous cells, which is fundamental during the formation of the esophagus and forestomach (PMID: 21068065).

Our data show a decreasing expression gradient of BMP ligands from the embryonic to adult stages, suggesting a potential involvement in GE-SCJ development. To clarify this finding, we have revised the text in the manuscript to: “*Further, the observed decreasing gradient in the expression of BMP pathway genes correlates with its regulatory roles in organ*

morphogenesis^{39,75}, however, its role in regulating GE-SCJ development needs further mechanistic evaluations.” Page 24 and Line 678-681.

- The assembly of the figures seems a bit rushed as quite some errors seem to be present. Please see my comments below. These require some work but are largely easily fixable.
- I am aware that the authors used some existing algorithms such as cell chat to analyze their data. However, the data representation is not clear. This concerns mainly Fig. 5b and c, and entire Fig. 6.

Response: We appreciate the reviewer's attention to detail, we have refined the figures mentioned.

• Figs. 5b and c:

It is not clear what is incoming or receiving? What does relative strength refer to?

Response: In our analysis, 'incoming' or 'receiver' signals refer to the communication received by a cell population through expressed receptors. Conversely, 'outgoing' or 'sender' signals pertain to the communication initiated by a cell population, typically through the expression of ligands.

The term 'relative strength' as used in these figures was calculated by normalizing the signaling strength of a pathway across epithelia and fibroblast cells for each individual time point (E19 and adult) in such a way that the values fall within the range 0-1. Relative strength calculation and visualization were done using the default functions provided by CellChat (PMID: 33597522).

As far as I understand if e.g. Notch is brown in adult epithelium and absent in fibroblast, it means it is expressed only in the epithelium. If it is labelled brown in E19 epithelium and fibroblast, it is produced in both tissue types

Response: In our heatmaps shown in Supplementary Fig.8b (previously Supplementary Fig. 5b), the color intensity, eg: for the Notch pathway, indicates the relative signaling strength. A less intense or no color in fibroblasts, suggests lower signaling strength compared to epithelial cells, rather than indicating a complete absence of the pathway. This information is provided in updated legend of Fig. 6b-c and Supplementary Fig. 8b-c.

How do I deduce now receiver and targeted cells based on these plots? In any case, common genes that are present in b and c should be located on the same height for the ease of the reader.

Response: In the updated Supplementary Fig. 8b-c (previously Supplementary Fig. 5b-c), we can infer that during both E19 and adult stages within the esophagus, the Notch pathway has a stronger incoming signal in the epithelium compared to fibroblasts, indicating that epithelial cells are the primary receivers of Notch signaling at both developmental stages. Conversely, fibroblasts show a notable outgoing Notch signal at the E19 stage, identifying them as senders during embryogenesis. At the adult stage, however, the epithelium emerges as the dominant sender of Notch signaling. More information regarding the cellular subtypes responsible for incoming or outgoing signaling within epithelia and fibroblasts can be deduced from Fig.7 and Supplementary Fig. 9 (previously Fig. 6 and Supplementary Fig. 6).

Further, to enhance readability and facilitate comparison, we have adjusted the heatmaps so that shared pathways between figures are uniformly positioned across Figures 6b-c and Supplementary Fig. 8b-c.

We have updated the manuscript text in the results section as “*Next, we identified the patterns for incoming, outgoing (Supplementary Fig. 8b-c), and overall signaling associated with epithelial and fibroblast cells (Fig. 6b-c). In our analysis, 'incoming' or 'receiver' signals refer to the communication received by a cell population through expressed receptors. Conversely, 'outgoing' or 'sender' signals pertain to the communication initiated by a cell population, typically through the expression of ligands. Our analysis indicated that fibroblasts predominantly served as the signaling senders during the epithelial-fibroblast interplay in the esophagus and stomach (Supplementary Fig. 8b-c). For Instance, in the esophagus, the Notch pathway has consistently stronger incoming signals in the epithelium compared to fibroblasts at both E19 and adult stages. At the E19 stage, fibroblasts predominantly exhibit outgoing Notch signals, whereas in adult tissues, epithelial cells emerge as the primary source. This pattern indicates that epithelial cells function as receivers of Notch signals across both examined stages. In contrast, fibroblasts transition from being predominant senders at E19 to a less active signaling role in adults (Supplementary Fig. 8b). This observation aligns with our earlier study, emphasizing the significance of basal squamous epithelial stem cells as the primary source of outgoing Notch signal and differentiated cells as the receivers contributing to stratification¹⁴.*” Page 15-16 and Line 456-472.

- Fig. 6a-g:

This figure is simpler as it shows the interpretation of the data obtained by the cell chat algorithm in a cartoon on the left and the expression data on the right. It is, however, not clear what exactly is plotted on the y-axis as it is not stated. Moreover, error bars are missing. Since the statement is that the ligands act on the same cells, it seems justified to make the same graphs for the corresponding receptors next to the ligand.

Response: To clarify, the y-axis in the trend plots represents the average gene expression levels as determined by scRNA-seq data calculated using the standard function 'Average Expression' from the R package Seurat.

In these plots, the calculated average expression value for each time point includes different cell types. As a result, the data distribution may be skewed, since certain cell types may not secrete certain ligands while others do. Thus, instead of plotting error bars directly on the graphs, we have now provided the complete data distribution statistics which includes standard deviation, standard error of mean, confidence interval 95% and quartile values for all ligands and receptors as a supplementary file (Supplementary Table 10) and referred in the main text.

Furthermore, we have added new trend plots that represent the expression of receptors in the signaling pathways. To avoid confusion and maintain clarity, these receptor plots are not directly juxtaposed with the ligand plots due to the complexity of ligand-receptor interactions, such as multiple ligands binding to the same receptor. However, circos plots in Fig. 7d-f and Supplementary Fig. 9d-f provide information on ligand-receptor specificity. All figures and their legends have been updated to reflect these changes.

- Fig. 6h-l:

While it may represent a good resource for the community, these figures are not interpretable. There are too many color codes. It is impossible identify the cell types of interest. These graphs need to be redone in simpler versions.

Response: We appreciate the reviewer's suggestion regarding the complexity presented by the multiple color codes. We have implemented a comprehensive revision of the color scheme to ensure clarity and ease of identification of cell types. The revised figures now employ a simplified color palette that distinguishes each cell type with a distinct shade:

Esophageal epithelial cells are uniformly represented in shades of 'orange'.

Fibroblasts from the esophagus are colored in shades of 'green'.

Stomach epithelial cells are designated in shades of 'blue'.

Stomach fibroblasts are identified by shades of 'magenta'.

These changes apply consistently across all relevant figures, ensuring that readers can more readily discern the cell populations of interest.

We have also updated the figure legends, specifically for Fig. 1c and 3a.

Minor criticisms:

- In the introduction the authors mentioned some literature references for scRNA seq data but left others out such as PMID: 34795 for Barrett's esophagus or PMID: 36717627 for gastric tissue.

Response: We agree with the reviewer's suggestion and have incorporated these references into our manuscript.

- Fig. 1d: For the embryonic cell cluster authors have calculated cell distribution from the different time points (Fig. 1r). Could they generate a similar plot also for the other clusters in the supplement. That could be informative to see how many specialized cells are already present at what developmental stages.

Response: As suggested by the reviewer, we have created additional plots to identify the presence of embryonic precursor cells in other clusters from esophagus and stomach samples at different time points, we generated cell distribution plots. These plots have been included in Supplementary Fig. 1e-g, along with feature plots that provide additional supporting evidence of co-expression levels of these precursor cell markers in UMAP space.

This information is now added in the results section as: *"Similarly, the precursor cell population was restricted to embryonic stages in the esophagus and stomach epithelia (Supplementary Fig. 1e-g)." Page 6 ; Line175-176*

- Fig 2k: Could authors clarify what happens to cells on the right side, which seem to get lost after embryonic stage E19. Are these E19 specific?

Response: The cell population depicted on the rightmost branch of Fig. 2i (Previously Fig. 2k) includes cells from E15, E19, and Pup developmental stages and is not exclusively associated with E19. These cells are characterized by a high expression of early embryonic markers such as *Vcan* and *Sox11*, coupled with a low expression or absence of differentiation markers like *Chga* and *Muc5ac*, marking them as an immature cohort with a unique transcriptional profile.

The presence of these cells across several stages suggests that they represent a transient, undifferentiated state in the developmental continuum.

As scRNA-seq data provides a snapshot of a cell's transcriptome at a given time, we can infer that these cells have not yet committed to differentiation stages into other cell types and likely progress towards differentiation, which may not be captured in the given dataset. To provide additional clarity, we have included new figures (Supplementary Fig. 4f-i) showcasing the expression patterns of the markers, which support our interpretation of cell's developmental status.

The figure legends and manuscript text have been updated to reflect this, in the results section as *“Whereas, in the stomach, we recovered a branching tree which clearly showed the ordering of cells from embryonic to adult time points with cells from base region confined separately from cells that belong to neck and pit regions (Fig. 2i). Additionally, in the rightmost branch of the trajectory, a combination of cells mostly from E15, E19 and few from pup time points exhibited expression of early embryonic markers like Sox11, Vcan, while differentiated cells such as Chga and Muc5ac were found in the left trajectories mainly in pup and adult states (Supplementary Fig. 4f-i). Since scRNA-seq data represents the cell's transient transcriptional state at a given time, it is inferred that the cells on the rightmost branch are in a precursor state, not yet committed to a definitive differentiation path, suggesting their potential to mature into specialized cell types as development proceeds.”*. Page 8, Lines 228-238

- Concerning organoid data:

- Mouse esophageal cultures were already previously published (PMID: 25373907). Please cite them.

Response: In accordance with this suggestion, we have now cited the relevant study (PMID: 25373907).

The text in the results section is updated as “Mouse esophageal stem cells grew into mature squamous stratified esophageal epithelial organoids in the presence and absence of WNT3a and RSPO1 (W/R) (Fig. 5a). However, they lost the stemness and growth capacity over a few passages in the presence of W/R (Fig. 5a, b, e, f). Consistently, patient-derived esophageal cells fail to form organoids in the presence of W/R, while their absence supports the growth and differentiation into mature stratified epithelium (Fig. 5c-d), in contrast to previous studies^{6,45}.” Page 8, Lines 344-349

- There is also already a publication about healthy human esophageal organoids (PMID: 34795059). Could you specify how your culture medium differs from them?

Response: Busslinger et al. (PMID: 33691112) utilized a specific cell culture medium consisting of 1% Noggin, B27 without vitamin A, 5mM nicotinamide, 10 μ M Forskolin (FSK), 100 ng/ml FGF10, 500 nM A83-01, 10% R-spondin conditioned medium, 25 ng/ml FGF7, 1 μ M P38 inhibitor, and Primocin. However, our experimental conditions differed in that we did not include R-spondin, FGF7, or the P38 inhibitor. Instead, we employed a medium containing 2 μ M TGF β inhibitor and 10 ng/mL EGF.

Interestingly, we observed that patient-derived esophageal cells were able to proliferate only when Wnt and R-spondin were absent from the culture medium (Fig. 5c). This stands in contrast to Busslinger et al.'s culture medium, which included R-spondin.

Also, see the updates mentioned in response to the previous comment.

- Establishing long term human esophageal cultures are of great interest as this is still challenging. Could authors provide data on how many passages human esophageal organoids could be maintained and how many different patients they have tried?

Response: In our study, we successfully initiated organoid cultures from stem cells enriched in a 2D environment from 5 different patients which were maintained for up to 4 passages before being biobanked. The organoids generated from these stem cells could be passaged 3-4 times using our esophageal medium.

Now we have included the data in the method section as *“Patient-derived esophageal stem cells were enriched and expanded in 2D from 5 patients for four passages before aliquoting and biobanking. We could generate 3D organoids from these stem cells that were passaged 3-4 times in the esophagus medium.”*. Page 24, Line 723-726-

- Page 11:

The authors mentioned they detected 34,393 genes. Generally, people refer with genes to protein coding genes. I assume they also included long non-coding RNAs etc. It would be great to specify that or restrict the analysis on protein-coding genes.

Response: Our data included protein-coding genes and long non-coding RNAs (lncRNAs). We have now specified this in the manuscript to ensure clarity: *“Microarray analysis revealed that among 34393 unique probes, encompassing protein-coding genes and long non-coding RNAs, 8030 genes were differentially regulated between columnar and squamous epithelium (Supplementary Fig. 7a, Supplementary Table 7).”* Page 12, Lines 375-378.

- Gene set enrichment analysis in Fig. 5a:

Some terminologies are highlighted but others are not. Can author explain what the others are and why only the selected ones were picked? In particular, since some entries have a similar expression pattern to the picked ones such as the two above 6.

Response: We have provided detailed data descriptions of all the enriched pathways together with their enrichment scores as a Supplementary Table 9. In the main figure, we selectively highlighted certain pathways based on their functional prominence and that showed significant differential enrichment between esophagus and stomach epithelia and fibroblasts at various time points.

- The authors mentioned couple of times the “panoramic view” of their data. I am not sure what they are referring to with this. The terminology does not seem very scientific.

Response: The phrase 'panoramic view' was metaphorically used to describe the broad and detailed scope of our data analysis, encompassing a wide array of cellular interactions and signaling patterns. However, recognizing that the term may not convey the intended scientific precision, we have revised the language in the manuscript for clarity.

- Some findings could be better put in context with the literature. E.g. Fgf7 is important for esophageal epithelium and Fgf10 for gastric one. This is also reflected in published organoid culture medium (gastric organoids need Fgf10: PMID: 25307862, PMID: 25539675, esophageal organoids grow with Fgf7: PMID: 34795059).

Response: We have incorporated the pertinent literature on FGF signaling within the esophageal and gastric epithelium, as suggested.

We have amended the text in the results section as “*The distinct expression profiles of FGF ligands in fibroblasts, with Fgf7 highly expressed in the esophagus and Fgf10 in the stomach, suggest a regulatory role in the GE-SCJ. The expression patterns of Fgf7 and Fgf10 align with their requirement for esophageal¹³ and stomach^{47,53} epithelium, as evidenced by organoid studies^{6,31}—nonetheless, their precise contribution to GE-SCJ development remains to be elucidated.*” page 17, Line 506-512

Comments to figure representation and suggestions:

- The color choices for the representation of scRNA seq data is often poorly chosen as it makes it difficult to follow their conclusion (e.g. Fig. 1c, 1l, 2i-l, 3a, S3a etc). It would be better stick to a color for either the same tissue and use different shades for the different time points. Alternatively use the same color for time points with different shades for the respective tissues.

Response: We have revised all our figures with a consistent color code—distinct colors for each tissue (esophagus, stomach) and cell type (epithelial, fibroblast) and different shades for various developmental stages, enhancing clarity and ease of interpretation across the manuscript.

Also, please see the above response to the related major comment on Fig. 6h-l.

- Fig. 1h: scale of y-axis is missing.

Response: We have added the scale to the y-axis in updated Fig. 1h.

- Fig. 1i, j, and k: Krt7 is highlighted in red in Fig. 1i and 1k but white in Fig. 1j whereas Krt8 is highlighted in red there. Please stick to the same color code which makes it easier for the reader.

Response: We have now standardized the color representation, ensuring that KRT7 is uniformly highlighted in red in the updated Fig. 1j and Supplementary Fig. 2c-e. Similarly, KRT8 is now consistently depicted in red in Fig. 1i and Supplementary Fig. 2a-b.

- Concerning all histological staining: please stay consistent with labeling. For mouse tissue use mouse nomenclature (Krt7) and not KRT7 whereas human sections should be labeled with the human nomenclature (KRT7).

Response: We have reviewed our figures and ensured that all labels conform to the standard nomenclature guidelines for mouse and human genes and proteins (PMID: 20685919).

All previous inconsistencies have been rectified to reflect this:

Mouse gene symbols are now correctly denoted in italics with only the first letter capitalized (e.g., *Krt7*). Mouse protein symbols are denoted with all capital letters (e.g., KRT7).

Human gene and protein symbols are both denoted with all capital letters (e.g., KRT7) and the gene symbols are additionally italicized (e.g., *KRT7*).

- Fig. 1i-k and 2a-c:

- The left panels are too small, and it is difficult to see anything in them. Please provide bigger versions of them.

Response: We have enlarged these images for better visibility and included them in Supplementary Fig. 2 and 3a-c. We have also reviewed and adjusted other figures throughout the manuscript to ensure that all images are clearly visible.

- For the ease of the reader, it would be great to label forestomach, columnar epithelium and esophageal remnants in the pictures.

Response: We have now labeled the relevant anatomical regions in the figures for clear identification, including 'Forestomach (Fs),' 'Hindstomach (Hs),' and 'Esophagus (Es).' Additionally, we have marked 'Sq' for squamous epithelia and 'Co' for columnar epithelia, specifically within the SCJ region, to aid the reader in distinguishing these key areas at a glance.

- Fig. 2l: The graphics is confusing. Some categories come up more often. Please replace it with something more understandable.

Response: To address this, we have revised the graphic to better convey the similarity between cell types across time points and tissues. We've implemented a consistent color-coding scheme that differentiates tissue types and developmental stages, applied uniformly across all figures.

- The legend of heatmaps is confusing:

- Fig. 2m, 4t-v, 5d, Supplementary Fig. 1b,d, 2c-d : the legend is “Avg. Expr “
- Fig. 2n: the legend is “TF Activity Score”
- Fig. 3e: the legend is “Expression”

I strongly assume that in each of these cases, the Z-score that is displayed. Please clarify this.

Response: Thank you for bringing this to our attention.

To clarify, all the heatmaps display scaled expression or activity values, which are indeed Z-scores. We have now standardized the legends across all relevant figures to reflect this, ensuring that 'Z-score' is clearly stated for consistency and ease of interpretation. The respective figures have been updated in the manuscript:

Fig. 2m (old) is now Fig. 2k (new).

Fig. 4t-v (old) is now Fig. 5t-v (new).

Fig. 5d (old) is now Fig. 6d (new).

Supplementary Fig. 1b,d (old) is now Supplementary Fig. 1b,c (new).

Supplementary Fig. 2c-d (old) is now Supplementary Fig. 4a-b (new).

Fig. 2n (old) is now Fig. 2l (new).

Fig. 3e (old) is now Fig. 3f (new).

- Fig. 3d, e: In Fig. 3d authors show that fibroblast clusters can be grouped into four subtypes but in Fig. 3e each cluster has a unique expression pattern. Could the authors clarify that and eventually provide a heatmap where the similarity can be seen. Moreover, it would be good to group the clusters in Fig. 3e according to the major subtypes in Fig. 3d.

Response: We have revised Fig. 3f (Previously Fig. 3e) to better illustrate the grouping of fibroblast subtypes and have created a new heatmap that showcases the similarities and differences in gene expression patterns within these groups. While individual clusters exhibit

unique expression profiles, our additional analysis has revealed a subset of genes that are consistently co-regulated within the major subtypes, which is now clearly depicted in the new heatmap. The figure and legend have been updated to reflect this accordingly.

- Fig. 3f and g are identical except for the highlighting of tissue type.

Response: Yes, Fig. 3g and h (Previously Fig. 3f-g,) are similar. This was intentional to delineate the distinct contributions of fibroblast populations from the esophagus and stomach. Given the complexity of the Sankey plots and the multitude of connections they represent, we opted to use separate plots specifically highlighting each tissue type.

- Fig. 3j: Is it really average exp value or percentage that is displayed? Anyway, could you please include error bars?

Response: Yes, y-axis represents the average exp value in Fig. 4c (Previously Fig. 3j).

Please see the response to the major comment on Fig. 6a-g. Similarly, now we have provided the complete data distribution statistics which includes standard-deviation, standard error of mean, confidence interval 95% and quartile values for all ligands and receptors as a Supplementary Table 6 and referred to in the main text.

- Fig. 5d:

The authors used a curated list for signaling pathways but unfortunately they did not group receptors or ligands of the same class next to each other, which seems to make more sense. E.g. Fzd receptors or Fgfrs are all over the place as well as Notch ligands.

Response: In figure 6d (Previously Fig. 5d), our objective was to construct a comprehensive ligand, receptor and mediators expression chart that provides valuable insights into identifying the cell types secreting specific ligands and expressing receptors. To achieve this, we carefully grouped the ligands, receptors and modulators to reveal their expression patterns illustrated in the dotplot.

Since some of the ligands interact with different receptors or receptor complexes within a signaling pathway, it is difficult to make a clear-cut grouping. Therefore, we choose to present common expression patterns of ligands, receptors, or modulators across epithelial and fibroblast subtypes to ease interpretation for readers. To make it easier for readers, we have color-coded the ligand, receptor, and modulator genes belonging to one pathway with the same color.

Conversely, Fig. 7 and Supplementary Fig. 9 (Previously Fig.6 and Supplementary Fig.6) focus on elucidating specific ligand-receptor combinations and their interactions.

Reviewer #2 (Remarks to the Author):

Throughout organogenesis, epithelial and the surrounding mesenchymal tissues highly communicate each other to induce complex and interconnected organs. Neighboring tissues secrete various paracrine factors to direct tissue growth and maturation of the organs. The gastroesophageal junction (GEJ) is defined as the junctional structure between esophagus and

stomach, which can be a hotspot of Barrett's esophagus. Therefore, the understanding of the process of GEJ development is important to clarify the mechanisms of metaplasia and cancer development. However, as described in manuscript, the process of GEJ development has not fully understood yet.

In this study, Kumar et al conducted single cell RNA sequence (scRNA-seq) analyses to investigate cell heterogeneity and signaling network governing the development of gastroesophageal junction (GEJ). The analyses of epithelial and mesenchymal lineage revealed that there is a diversity in gastroesophageal cells during the development. They further showed different Wnt dependency between esophageal and gastric epithelium by taking advantage of both mouse model and organoid technologies. Finally, the study has provided signaling interplay between epithelium and fibroblasts at GEJ. Overall, the results are informative and interesting for the readers, especially in the field of organogenesis, of nature communications. However, the data should be more spatially validated with staining of respective markers. The authors are also encouraged to address several concerns as written below.

Response: Thank you very much for appreciating the importance of our work and the feedback. We have addressed your concerns in the following detailed responses.

- scRNAseq analyses revealed the diversity of cell types during GEJ development (Sqs, FBs). However, the distribution of these subtypes is unclear. The authors should perform immunostaining or in situ hybridization for the respective markers to provide spatial information of these cell types.

Response: Thank you for recommending additional spatial analyses. We have expanded our immunostaining and smRNA-ISH studies in addition to some of the spatial analysis that was presented already.

1. Precursor cells of epithelia: We located *Sox11* and *Cldn6* positive cells at E15, and showed their absence in adult tissues, and associated them with markers for squamous (KRT5, P63) and columnar (KRT7, KRT8) epithelia (Fig. 1i-j, 2a-c; Supplementary Fig. 2, 3a-e).
2. Transcription Factors (TF): SOX2 and GATA6 expression was spatially validated, correlating with their distinct distribution in squamous and columnar epithelium as per TF activity analysis (Fig. 2m; Supplementary Fig. 4j).
3. Epithelial Differentiation Markers: Different states of differentiated squamous epithelial cells expressing LOR, KRT6, and JUN were identified alongside basal and parabasal cell markers (P63, KRT17). CHGA+ and MUC5AC+ differentiated cells were located among KRT7+ cells, with *Lgr5+* and *Axin2+* marking stem cells in columnar epithelium (Fig. 4o; 5l, g, w-x; Supplementary Fig. 4 c-d; 6c-e)
4. Stromal Cell Markers: We have demonstrated the distinct expression patterns of stromal cell markers ACTA2 and POSTN, as well as the epithelial marker CDH1 (Fig. 3e; Supplementary Fig. 6a-b). These markers clearly define the unique organization of stromal subtypes beneath the epithelial layers, providing a clear visualization of the diverse cellular architecture within the esophageal and stomach regions.

5. Signaling Pathways: The expression of Wnt pathway regulators like *Rspo3*, *Dkk2*, and *Sfrp4* was demonstrated in distinct stromal cells, visualized through co-staining with ACTA2 and POSTN (Fig. 4d, g, j-l). For e.g., POSTN low fibroblast just below the stomach epithelial gland expresses *Rspo3* (Fig. 4j lower panel). While *Sfrp4* is mainly expressed in the esophagus that is distinct from ACTA2+ cells (Fig. 4l upper panel).
6. PDGF Pathway: We showed that *Pdgfa* is expressed in squamous and columnar epithelium in the basal cells, while its receptor PDGFRA is expressed in the underlying fibroblast (Fig. 7g-h).

- Based on scRNA seq and cell-cell communication analyses, the authors suggest ligand-receptor interaction between epithelial cell types and fibroblast cell types during GEJ development. However, these analyses do not demonstrate that sender cells and receiver cells are located closely enough to signal each other. It would be better to stain the ligand-receptor pairs, and target genes of signaling pathways.

Response: To demonstrate the proximity of signal sender and receiver cells, we chose the PDGF signaling pathway as the candidate as it showed a unique signaling interaction pattern from epithelial cell to fibroblast as shown in Fig. 7c. In line with this, a previous study showed that PDGFA expression in intestinal epithelium signals with PDGFRA expressing stromal cells for proper Villi formation during intestinal development (PMID: 10903171).

Our ligand-receptor interaction predictions from single-cell data showed that ligand *Pdgfa* is primarily sent by Sq1-2 of the esophagus or tuft/endocrine cell types of the stomach and received by different fibroblasts (Fig. 7f). We have performed smRNA-ISH combined with immunofluorescence to show the *Pdgfa* gene expression from the epithelial stem cell compartment of the adult esophagus and stomach, and its receptor PDGFRA expression in the fibroblasts. This result clearly showed the proximity of sender cells (epithelia) and receiver cells (fibroblast) in both the esophagus and stomach, suggesting possible interactions (Fig. 7g-h). Nevertheless, since PDGFA is a morphogen, it might also induce a gradient of signaling to the distant receptor-expressing cells.

This information is updated in the results section as “*Further, we spatially validated one of the key L-R interaction predictions where the Pdgfa ligand is primarily sent by Sq1-2 of the esophagus and tuft/endocrine cell types of the stomach targeting different fibroblasts (Fig 7f). We confirmed the proximity of Pdgfa sender cells (epithelia) and PDGFRA-expressing receiver cells (fibroblast) in both the esophagus and stomach, suggesting possible interaction (Fig 7g-h). In line with this, a previous study showed that PDGFA expressing intestinal epithelium signals with PDGFRA expressing stromal cells for proper villi formation during gastrointestinal development*⁵⁹.” Page18, Line 570-579.

-The study shows that Wnt signaling is activated in esophageal compartment but inactivated in stomach. Fig. S6 shows ligand-receptor pairs of Wnt signaling, but authors did not mention these in result sections. Authors should describe what Wnt ligands, receptors and antagonists in addition to *Rspo3* and *Dkk2* are expressing at GEJ.

Response: We would like to clarify that Wnt signaling is indispensable for the regeneration of stomach epithelial stem cells, while its inhibition is supportive of esophageal stem cells. These findings have been demonstrated in Fig. 4 and 5.

As per the findings, Fig. 6d provides a detailed summary of the expression profiles of Wnt ligands and receptors in different epithelial and fibroblast cell populations. In the esophagus, it was observed that Wnt receptor *Fzd6* was prominently expressed by epithelial cells, while *Fzd2* was primarily expressed by underlying fibroblasts. Among the Wnt ligands, *Wnt4*, *Wnt10a*, *Wnt5b*, and *Wnt7b* were found to be enriched in the epithelial compartment, while *Wnt5a* was predominantly expressed in fibroblasts. Moreover, the esophagus had an abundance of Wnt inhibitors, such as *Sfrp4* and *Dkk2*, particularly in the stromal region (Fig. 4 g,k-l; Fig. 6d). In addition, the DKK2 receptor *Kremen1* was expressed in epithelial cells (Fig. 4m; Fig. 6d). It should be noted that DKK2 is known to interact with the Kremen1 protein, leading to the internalization of LRP6 and subsequent inhibition of canonical Wnt signaling, as demonstrated in previous studies (PMID: 12050670, PMID: 17143291). Fibroblast-secreted SFRP4 serves as a soluble decoy receptor for Wnt ligands, thereby antagonizing both canonical and non-canonical Wnt/ β -catenin pathways, as supported by previous studies (PMID: 18322270). Additionally, the high expression of *Fzd6*, an inhibitor of canonical Wnt signaling (PMID: 14747478), in esophageal epithelia, suggests a role for Wnt signaling inhibition in esophageal development.

In the stomach, higher levels of Wnt receptors *Lrp5*, *Lrp6*, and *Fzd5* are expressed in epithelia, while *Fzd1* and *Fzd2* are expressed in fibroblasts, and *Fzd4* in both epithelial and fibroblast cells. The Wnt ligand *Wnt4* is expressed in the epithelial compartment, while *Wnt5a* is predominantly expressed in fibroblasts. Additionally, higher levels of *Rspo3* is expressed in fibroblasts (Fig. 4d, j; Fig. 6d). We have also shown that the RSPO3 receptor *Lgr5*, which is necessary to potentiate canonical Wnt signaling, is specifically expressed in the epithelial stem cells of the stomach but not in the esophagus epithelium (Supplementary Fig. 6d).

Our analysis of Wnt signaling ligand-receptor pairs revealed that esophageal cells express *Wnt4*, *Wnt10a*, *Wnt7b*, *Wnt5a*, and *Wnt11* ligands (In Supplementary Fig. 9f,), which are involved in either one or both canonical and non-canonical Wnt pathways. Interestingly, we found that the majority of Wnt signal senders are epithelial cells, and receivers are fibroblasts, while non-canonical *Wnt5a* and *Wnt11* signals are primarily exchanged among fibroblasts. On the other hand, in the stomach, the senders and receivers bi-directionally communicate between the epithelial and fibroblast compartments.

We have updated the text in the results section as “*Our ligand-receptor analysis of WNT signaling revealed that esophageal cells express Wnt4, Wnt10a, Wnt7b, Wnt5a, and Wnt11 ligands (Supplementary Fig. 9f) involved in either one or both canonical and non-canonical WNT pathways. Interestingly, most WNT signal senders were epithelial cells, and receivers were fibroblasts, while non-canonical Wnt5a and -Wnt11 signals were primarily restricted to senders and receivers within fibroblasts. On the other hand, in the stomach, Wnt4 and Wnt5a gene expression were observed, with senders and receivers being bi-directional between epithelial and fibroblast compartments (Supplementary Fig. 9f).* “. Page18, Line 563-570

-Fig. 3k, n: Cell types are not clear. Marker genes (e.g. **SMA**, TAGLN) should be co-stained to show the cell types which express *Rspo3* and *Dkk2*.

Response: We performed smRNA-ISH staining for *Rspo3* and *Dkk2* gene in combination with immunofluorescence staining for fibroblast and epithelial cell type markers POSTN, and CDH1 respectively. The images were added as Fig. 4j-k.

-Fig. 3s: It is difficult for the readers to see which tissue is activated in Wnt signaling. The authors should perform double or triple co-staining for Axin2-lineage with epithelial and/or mesenchymal e markers.

Response: The previous publication (PMID: 28813421) has demonstrated AXIN2 lineage tracing within stomach epithelia, which aligns with our current observation that AXIN2+ cells are specifically marked at the basal region of the glandular epithelium (Fig. 4o, Supplementary Fig. 6c, e). However, a novel finding in our study is the absence of AXIN2 lineage tracing in esophageal epithelial cells co-immunostained for KRT5, suggesting that AXIN2+ cells do not contribute to the population of squamous cell types in the esophagus. To provide a comprehensive view of this, we have included a larger section of the GE-SCJ with AXIN2 lineage-marked cells highlighted in red and KRT5 in green (Supplementary Fig. 6c).

-Fig. 6: The data of signaling networks are too complex at first glance. In order to summarize critical points, table and/or cartoons would be helpful for the readers.

Response: We thank the reviewer for the valuable suggestion to highlight crucial components that enhance comprehension. We have now generated a cartoon (Fig. 8), that summarizes signaling information across different compartments at different developmental time points.

REVIEWER COMMENTS

Reviewer #1 (Remarks to the Author):

The manuscript improved in overall quality. The authors addressed many of my previous comments and I thank them for that. However, there are still some open questions.

1.

The authors provided Sox11 and Cldn6 staining in the revised manuscript, but I am not sure how well these staining are fitting into the bioinformatic analysis. In the manuscript, it is mentioned that Sox11 and Cldn6 are embryonic epithelial cell markers, and these cells commit then to either squamous or columnar lineages. Based on the provided staining, it seems that these two genes are marking only columnar and not squamous cell epithelium in E15 embryos. Can the author comment on the discrepancy to the bioinformatic analysis? To what degree does the bioinformatic conclusion reflect the in vivo situation?

2.

Later in the manuscript the authors argue that transcription factors such as Nanog, Tead1, Prdm14 or Pax5 may play a role for lineage commitment. They state that their mechanistic role is still unclear and is the basis for future research. I consider it however important to validate this statement with histological staining for at least one of these marker genes.

3.

Previous comment:

Figs. 5b and c:

It is not clear what is incoming or receiving? What does relative strength refer to?

Response: In our analysis, 'incoming' or 'receiver' signals refer to the communication received by a cell population through expressed receptors. Conversely, 'outgoing' or 'sender' signals pertain to the communication initiated by a cell population, typically through the expression of ligands.

The term 'relative strength' as used in these figures was calculated by normalizing the signaling strength of a pathway across epithelia and fibroblast cells for each individual time point (E19 and adult) in such a way that the values fall within the range 0-1. Relative strength calculation and visualization were done using the default functions provided by CellChat (PMID: 33597522).

Previous comment:

As far as I understand if e.g. Notch is brown in adult epithelium and absent in fibroblast, it means it is expressed only in the epithelium. If it is labelled brown in E19 epithelium and fibroblast, it is produced in both tissue types.

Response: In our heatmaps shown in Supplementary Fig.8b (previously Supplementary Fig. 5b), the color intensity, e.g.: for the Notch pathway, indicates the relative signaling strength. A less intense or no color

in fibroblasts, suggests lower signaling strength compared to epithelial cells, rather than indicating a complete absence of the pathway. This information is provided in updated legend of Fig. 6b-c and Supplementary Fig. 8b-c.

New comment:

I understand that a pre-existing R package was used from another publication. Nonetheless, it merits to have a qualitative explanation in the text for the reader. It is still hard to understand even with the provided explanation in the rebuttal letter. It would be great to see the data behind it for the examples that are discussed in the text. Generally the discussion about the incoming and outgoing signal analysis feels very extensive in the manuscript although it is only based on few computational graphs.

4.

Figure 8 d-f are still very confusing. As far as I understand the signaling cells are all the bottom and signal up to the top half. If I am not mistaken, the standard chord diagram in CellChat does not make this separation. In the current representation, certain cell types can thus be present twice (at the top and at the bottom), if they signal to themselves. This is quite confusing. If this representation should be kept, it would make more sense to provide the cell type name instead of BMP2/4 and BR1 etc. and color code the arrow with the signaling interactions. The current color code of the arrows is not clear.

5.

Now that z-scores in the figures are clearer, the question arises for the reader what transcriptional activity is. This is based on a R package. Within the text this comes out of the blue and it is not really explained what it means without reading up on the other paper. This should be fixed for the convenience of the reader.

6.

Previous comment:

- Fig 2k: Could authors clarify what happens to cells on the right side, which seem to get lost after embryonic stage E19. Are these E19 specific?

Response: The cell population depicted on the rightmost branch of Fig. 2i (Previously Fig. 2k) includes cells from E15, E19, and Pup developmental stages and is not exclusively associated with E19. These cells are characterized by a high expression of early embryonic markers such as Vcan and Sox11, coupled with a low expression or absence of differentiation markers like Chga and Muc5ac, marking them as an immature cohort with a unique transcriptional profile. The presence of these cells across several stages suggests that they represent a transient, undifferentiated state in the developmental continuum.

As scRNA-seq data provides a snapshot of a cell's transcriptome at a given time, we can infer that these cells have not yet committed to differentiation stages into other cell types and likely progress towards differentiation, which may not be captured in the given dataset. To provide additional clarity, we have included new figures (Supplementary Fig. 4f-i) showcasing the expression patterns of the markers, which support our interpretation of cell's developmental status.

New comment:

It is still not clear what the take home message of Fig. 2i. There is a bunch of cells that cease to exist in the adult stage. Interestingly, the disappearing cells at E19 stage are assigned specific cell types in the plot such as neck, pit etc. I am struggling with the biological meaning, even when provided with the response from the authors.

7.

Page 11: "Consistently, patient-derived esophageal cells fail to form organoids in the presence of W/R, ..., in contrast to previous studies 6,45".

This statement is not correct. Ref 6 did not use WNT in the culture medium, only Rspo whereas the ref 45 used WNT and Rspo. Please correct.

8.

Fig. 2j: It is still not clear what the biological interpretation of this dendrogram should be.

9.

Fig. 7 g and h:

How is proximity defined? Could authors provide a quantification?

Reviewer #2 (Remarks to the Author):

I would thank the authors for their large efforts to address most of my concerns, including spatial experiments for marker genes, and signaling molecules. A considerable number of additional experiments improved the manuscript. Also, the graphical summary in Fig. 8 will help emphasize the highlights of the study and help readers understand the overview of the paper.

Minor comments

1. Abstract: The authors focus too much on GE-SCJ. Also, the abstract sound like this study demonstrates the fibroblast-epithelial interaction at the GE-SCJ. However, strictly speaking, the authors analyzed esophageal and gastric compartments but not GE-SCJ itself.
2. Line 283: The number of Figure is missing.
3. Materials and Methods: The catalog number should be provided for all reagents.
4. Materials and Methods: Esophageal organoids and Stomach organoid sections are poorly explained. The authors should provide more detail information (eg, tissue numbers, cell numbers, reagent volume, etc...) so that the work can be reproduced.

Reviewer #1 (Remarks to the Author): The manuscript improved in overall quality. The authors addressed many of my previous comments and I thank them for that. However, there are still some open questions.

Response: Thank you very much for your appreciation of the revised manuscript.

1.

The authors provided Sox11 and Cldn6 staining in the revised manuscript, but I am not sure how well these staining are fitting into the bioinformatic analysis. In the manuscript, it is mentioned that Sox11 and Cldn6 are embryonic epithelial cell markers, and these cells commit then to either squamous or columnar lineages. Based on the provided staining, it seems that these two genes are marking only columnar and not squamous cell epithelium in E15 embryos. Can the author comment on the discrepancy to the bioinformatic analysis? To what degree does the bioinformatic conclusion reflect the in vivo situation?

Response:

(1) In the manuscript (page 5; Line 129), we stated, 'Differential expression analysis across GE-SCJ epithelial cell clusters unraveled the gene expression signature associated with the process of embryonic epithelial cells (*Sox11*, *Igf2*, *H19*, *Cldn6*, *Vcan*, and *Bex1*)...'. Since we intended to convey that a specific gene signature, including *Sox11* and *Cldn6*, characterizes embryonic 'precursor' epithelial cell populations (columnar type), we have modified the text to enhance clarity for readers as follows:

"Differential expression analysis across GE-SCJ epithelial cell clusters unraveled the gene expression signature associated with embryonic precursor epithelial cells (*Sox11*, *Igf2*, *H19*, *Cldn6*, *Vcan*, and *Bex1*)..."

(2) In aligning our bioinformatics with in vivo findings, smRNA-ISH in Suppl. Figure 2b confirms strong Cldn6 expression in E15 epithelial cells at the gastroesophageal junction and stomach, matching the trends in our scRNA-seq data (Figure 1h). Additionally, we observe high Sox11 expression in embryonic precursor epithelial cells, which diminishes in later stages (E19 to adulthood), as illustrated in Suppl. Fig. 2a and Fig. 1i and h. In Fig. 1i and Suppl. Fig. 2a, the transition toward the squamous (Sq) lineage is discernible to the left of the precursor cells. Given the developmental stage at E15, precise boundary delineation is challenging due to the evolving spatial organization of squamous and columnar cells at the GE-SCJ. Recognizing that initial boundary lines previously drawn at E15 were overly simplistic, we have now replaced the demarcating line with a more representative depiction of the precursor cell region in our revised figures.

2.

Later in the manuscript the authors argue that transcription factors such as Nanog, Tead1, Prdm14 or Pax5 may play a role for lineage commitment. They state that their mechanistic role is still unclear and is the basis for future research. I consider it however important to validate this statement with histological staining for at least one of these marker genes.

Response: In our revised manuscript, we stated on Page 9, Line 250:

"Further, TFs such as *Nanog*, *Tead1*, *Prdm14*, *Pax5* activity were enriched in the early-stage squamous epithelium and specific cell states of columnar epithelia (Fig. 2l, and Suppl. Table 4)."

This observation was intended to highlight the potential involvement of these TFs in epithelial differentiation rather than definitively asserting their role in lineage commitment. We further

clarified this in the manuscript: "However, their mechanistic role in lineage commitment within the squamous and columnar epithelia is unclear and an avenue for future research."

To support our single-cell RNA sequencing data, we have already conducted spatial validations for several transcription factors, including p63, Sox11, Cldn6, GATA6, and SOX2. These validations illustrate the distinct expression patterns and spatial distribution of these TFs across different epithelial types, potentially indicating their roles in lineage specification. The complete list of these transcription factors and their expression patterns can be found in Table 4. We acknowledge the importance of histological validations for all the other mentioned transcription factors (such as Nanog, Tead1, Prdm14, Pax5, and others from Table 4). However, we must also consider the scope and resources available for this study. The process of generating additional staining for each of these transcription factors would be extensive and, we believe, is beyond the immediate scope of this manuscript.

Thus, our current manuscript provides a foundation for understanding the potential involvement of these additional TFs in epithelial differentiation, with the caveat that their definitive roles remain to be elucidated in future studies.

3.Previous comment:

Figs. 5b and c:

It is not clear what is incoming or receiving? What does relative strength refer to?

Response: In our analysis, 'incoming' or 'receiver' signals refer to the communication received by a cell population through expressed receptors. Conversely, 'outgoing' or 'sender' signals pertain to the communication initiated by a cell population, typically through the expression of ligands.

The term 'relative strength' as used in these figures was calculated by normalizing the signaling strength of a pathway across epithelia and fibroblast cells for each individual time point (E19 and adult) in such a way that the values fall within the range 0-1. Relative strength calculation and visualization were done using the default functions provided by CellChat (PMID: 33597522).

Previous comment:

As far as I understand if e.g. Notch is brown in adult epithelium and absent in fibroblast, it means it is expressed only in the epithelium. If it is labelled brown in E19 epithelium and fibroblast, it is produced in both tissue types.

Response: In our heatmaps shown in Supplementary Fig.8b (previously Supplementary Fig. 5b), the color intensity, e.g.: for the Notch pathway, indicates the relative signaling strength. A less intense or no color in fibroblasts, suggests lower signaling strength compared to epithelial cells, rather than indicating a complete absence of the pathway. This information is provided in updated legend of Fig. 6b-c and Supplementary Fig. 8b-c.

New comment: I understand that a pre-existing R package was used from another publication. Nonetheless, it merits to have a qualitative explanation in the text for the reader. It is still hard to understand even with the provided explanation in the rebuttal letter. It would be great to see the data behind it for the examples that are discussed in the text. Generally the discussion about the incoming and outgoing signal analysis feels very extensive in the manuscript although it is only based on few computational graphs.

Explanatory Figure. 1

Response: Using the above explanatory figure 1, we would like to illustrate the interpretation of outgoing and incoming signaling data generated for the reviewer, using NOTCH signaling as an example since we have discussed this in (Page 14; Line 439) the manuscript (Suppl. Fig. 8b). Zooming into the esophagus signaling patterns in Suppl. Fig. 8b, the heatmap shows strong incoming signaling patterns in epithelia at E19 and adult time points. Outgoing signals predominantly originate from fibroblast cells during E19, shifting to epithelia during the adult stage (Explanatory Fig.1a).

Since the signaling patterns derived for the NOTCH pathway are based on several ligand-receptor pairs, we first extracted the list of L-R pairs involved in establishing this signaling and their overall contribution (Explanatory Fig.1b). Notably, the top hits: Dlk1-Notch1 and Jag2-Notch1 pairs, were the highest contributors for NOTCH signaling during E19 and adult time points, respectively.

Next, by visualizing the CellChat-predicted interactions between different epithelial and fibroblast subclusters for these top L-R pairs, it is evident that epithelial subpopulations express the Notch receptor Notch1 at both time points and receive the ligands Dlk1 and Jag1. This confirms the observed strong incoming signaling pattern in epithelia (Explanatory Fig.1c-d).

In Explanatory Fig.1d, we could see a specific interaction where fibroblast subcluster FB5 sends the ligand Jag1 to epithelial cells. Despite fibroblasts contributing signals, their strength is lower when compared to cumulative signals sent by the epithelial cells. This is a clear example of

relative strength scaling, as reflected in the final heatmap, where the fibroblast section exhibits less intense or no color in the adult esophagus.

Thus, these heatmaps present a broad insight into overall signaling participation between epithelia and fibroblasts over time. It is important to consider the broad nature of the patterns obtained, which encompass different subclusters within each cell population and various L-R pairs involved in signaling. Therefore, the presented heatmaps provide summarized communication probabilities of all ligand-receptor interactions associated with each signaling pathway, offering insights into the complex cellular communication network over time. For detailed information regarding CellChat functionalities and statistical methods, the readers are referred to the original article PMID: 33597522.

4. Figure 8 d-f are still very confusing. As far as I understand the signaling cells are all the bottom and signal up to the top half. If I am not mistaken, the standard chord diagram in CellChat does not make this separation. In the current representation, certain cell types can thus be present twice (at the top and at the bottom), if they signal to themselves. This is quite confusing. If this representation should be kept, it would make more sense to provide the cell type name instead of BMP2/4 and BR1 etc. and color code the arrow with the signaling interactions. The current color code of the arrows is not clear.

Response: In understanding the complexity of interpreting cell-cell signaling interactions, we aimed for a balance between clarity and comprehensive data representation. In our original circos diagrams, the bottom half represents the signal-sending cell groups, while the top half shows the signal-receiving groups. This layout is generated using the 'netVisual_chord_gene' function from CellChat, designed to illustrate ligand-receptor signaling interactions. The color coding of arrows is based on the signal-sending group, as shown in Explanatory Fig. 2a, which would facilitate a better understanding of the signaling dynamics.

It is important to note that the inclusion/occurrence of the same cell types in both the bottom and top halves serves a specific purpose as it emphasizes both autocrine and paracrine signaling interactions.

However, considering the reviewer's feedback, we experimented with an alternative representation (Explanatory Fig.2b-c), where arrows are color-coded according to signaling interactions, and cell names are labeled instead of ligand-receptor pairs.

However, upon careful consideration, we found that this alternative approach poses several challenges:

1. It becomes more difficult to distinguish between autocrine and paracrine signaling, an essential aspect of our analysis.
2. Adding additional color codes to represent cell types and interactions for various ligand-receptor pairs from different signaling pathways adds to the visual complexity.
3. Tracking the signaling source becomes challenging with the abundance of chords in the diagram. For instance, in Explanatory Fig. 2c, both pit and fibroblast cells (S_FB_6) express the ligand Bmp2 (in shades of yellow). While S_FB_2 receives these signals, tracing the signaling arrows back to their origin (whether sent by Pit cells or S_FB_6 cells) is problematic due to the numerous chords in the visuals, as highlighted in Explanatory Fig. 2c.

Explanatory Figure. 2

Considering these challenges and the importance of conveying the key information effectively, we have decided to retain our previously revised form of representation.

5. Now that z-scores in the figures are clearer, the question arises for the reader what transcriptional activity is. This is based on a R package. Within the text this comes out of the blue and it is not really explained what it means without reading up on the other paper. This should be fixed for the convenience of the reader.

Response: Thank you for your feedback. Here, TF activity is calculated computationally using the gene expression of their targets.

This information is updated in the main text of the manuscript on Page 8, Line 235: ‘We computed TF activities based on the expression levels of their target genes. TF-target interactions were sourced from curated evidence with high confidence levels using DoRoThEA⁸⁷. This revealed an overlap of cell cycle-related genes....’

6. Previous comment: - Fig 2k: Could authors clarify what happens to cells on the right side, which seem to get lost after embryonic stage E19. Are these E19 specific?

Response: The cell population depicted on the rightmost branch of Fig. 2i (Previously Fig. 2k) includes cells from E15, E19, and Pup developmental stages and is not exclusively associated with E19. These cells are characterized by a high expression of early embryonic markers such as *Vcan* and *Sox11*, coupled with a low expression or absence of differentiation markers like *Chga* and *Muc5ac*, marking them as an immature cohort with a unique transcriptional profile. The presence of these cells across several stages suggests that they represent a transient, undifferentiated state in the developmental continuum.

As scRNA-seq data provides a snapshot of a cell's transcriptome at a given time, we can infer that these cells have not yet committed to differentiation stages into other cell types and likely progress towards differentiation, which may not be captured in the given dataset. To provide additional clarity, we have included new figures (Supplementary Fig. 4f-i) showcasing the expression patterns of the markers, which support our interpretation of cell's developmental status.

New comment:

It is still not clear what the take home message of Fig. 2i. There is a bunch of cells that cease to exist in the adult stage. Interestingly, the disappearing cells at E19 stage are assigned specific cell types in the plot such as neck, pit etc. I am struggling with the biological meaning, even when provided with the response from the authors.

Response: The rightmost branch in Fig. 2i exhibits markers characteristic of neck-like and pit-like cell types. As illustrated in Figure 2j, these cells are transcriptionally distinct from the fully differentiated adult Neck/Pit cells. These embryonic differentiated cells could be in a transient state and may differentiate to the adult type or likely shed off as development progresses.

To improve clarity, the previous information on Page 8, Line-204: “Given that scRNA-seq data represents the cells transient transcriptional state at a given time, it is inferred that the cells on the rightmost branch are in a precursor state, not yet committed to a definitive differentiation path, suggesting their potential to mature into specialized cell types as development proceeds.” has been modified as follows in the updated manuscript Page 8 and Line 215:

“Since scRNA-seq data represents the cell's transcriptome at a given time, it is inferred that the embryonic differentiated cells (neck-like and pit-like), which are distinct from the differentiated adult cells on the rightmost branch, could indicate transient states and may differentiate to the adult type or likely shed off during development.”

7. Page 11: “Consistently, patient-derived esophageal cells fail to form organoids in the presence of W/R, ..., in contrast to previous studies 6,45”. This statement is not correct. Ref 6 did not use WNT in the culture medium, only Rspo whereas the ref 45 used WNT and Rspo. Please correct.

Response: The sentence is modified and updated in the revised manuscript as below: “Consistently, patient-derived esophageal cells fail to form organoids in the presence of W/R, while their absence supports the growth and differentiation into mature stratified epithelium (Fig. 5c-d). This is in contrast to previous studies that showed the culture of esophageal organoids with either the Wnt agonist R-Spondin alone (6) or in combination with a Wnt ligand (45), suggesting Wnt signal is dispensable for the esophageal organoid formation.” (Page 11; Line 331)

8. Fig. 2j: It is still not clear what the biological interpretation of this dendrogram should be.

Response: The dendrogram in Fig. 2j provides insights into the distinct transcriptional profiles between individual cell clusters and their spatial distribution across the UMAP space. Notably, it highlights three primary branches, delineating (1) esophagus squamous epithelia across all time points, (2) columnar epithelia from embryonic to pup stage, and (3) adult stomach epithelia (please refer to the figure above). These main branches are further subdivided into smaller groups, revealing their relationships in gene expression space.

Now, this interpretation is included for the readers in the updated manuscript as follows on Page 8, Line 223.

“In the esophagus, basal and parabasal cells occupy separate subbranches, while highly differentiated cells (Sq2C-Pup and Sq3-Adult) appeared in a distinct subbranch, revealing transcriptional distinction between these cell types. Similarly, in the stomach, epithelial cells formed a separate branch from the adult time point, emphasizing the well-developed glandular units comprising complex cell types distinct from earlier developmental time points.”

9. Fig. 7 g and h:

How is proximity defined? Could authors provide a quantification?

Response: The data presented in Fig 7g-h clearly shows the spatial vicinity of the ligand and receptor-expressing cells neighboring (proximal) each other. It is well established that ligands are secreted (e.g., PDGFA) from the cells that express them and diffuse to signal to their neighboring cells that express corresponding receptors.

Thus, similar to previous studies demonstrating spatial validation of Ligand-receptor expression of the predicted cell-cell communication (PMID: 33597522), the presented data indicate the high probability of interaction between the *Pdgfa* sender cells (epithelial) and *PDGFRA*-expressing receiver cells.

Reviewer #2 (Remarks to the Author):

I would thank the authors for their large efforts to address most of my concerns, including spatial experiments for marker genes, and signaling molecules. A considerable number of additional experiments improved the manuscript. Also, the graphical summary in Fig. 8 will help emphasize the highlights of the study and help readers understand the overview of the paper.

Response: Thank you very much for recognizing our efforts and your appreciation of the revised manuscript.

Minor comments

1. Abstract: The authors focus too much on GE-SCJ. Also, the abstract sound like this study demonstrates the fibroblast-epithelial interaction at the GE-SCJ. However, strictly speaking, the authors analyzed esophageal and gastric compartments but not GE-SCJ itself.

Response: We would like to highlight that our investigation aimed to understand the establishment and homeostasis of the GE-SCJ during development. Employing methodologies such as immunohistochemistry (IHC), single-molecule RNA in situ hybridization (smRNA ISH), and single-cell sequencing, our study meticulously examined the expression patterns and spatial distribution of specific markers across the esophagus, stomach, and the GE-SCJ. This approach enabled us to characterize the unique cellular environments within the gastric and esophageal areas and their convergence at the GE-SCJ.

Considering the reviewer's feedback, we have revised the abstract: “We identified distinct transcriptional states and signaling pathways in the epithelial and mesenchymal compartments of the esophagus and stomach during development.” Page 2; Line 40.

2. Line 283: The number of Figure is missing.

Response: Please excuse the oversight. We should have deleted the “Fig” in the previous version. Now it is removed.

3. Materials and Methods: The catalog number should be provided for all reagents.

Response: Now we have included these details.

4. Materials and Methods: Esophageal organoids and Stomach organoid sections are poorly explained. The authors should provide more detail information (eg, tissue numbers, cell numbers, reagent volume, etc...) so that the work can be reproduced.

Response: We have included a more elaborate protocol.

REVIEWERS' COMMENTS

Reviewer #1 (Remarks to the Author):

Thank you very much for the clarifications.

I do not have any further comments.